**JCB** Journal of Cell Biology

# High sugar diet–induced fatty acid oxidation potentiates cytokine-dependent cardiac ECM remodeling

Jayati Gera[1], Dheeraj Kumar[1]*, Gunjan Chauhan[1]*, Adarsh Choudhary[1], Lavi Rani[1], Lolitika Mandal[2], and Sudip Mandal[1]

Context-dependent physiological remodeling of the extracellular matrix (ECM) is essential for development and organ homeostasis. On the other hand, consumption of high-caloric diet leverages ECM remodeling to create pathological conditions that impede the functionality of different organs, including the heart. However, the mechanistic basis of high caloric diet–induced ECM remodeling has yet to be elucidated. Employing in vivo molecular genetic analyses in *Drosophila*, we demonstrate that high dietary sugar triggers ROS-independent activation of JNK signaling to promote fatty acid oxidation (FAO) in the pericardial cells (nephrocytes). An elevated level of FAO, in turn, induces histone acetylation–dependent transcriptional upregulation of the cytokine Unpaired 3 (Upd3). Release of pericardial Upd3 augments fat body-specific expression of the cardiac ECM protein Pericardin, leading to progressive cardiac fibrosis. Importantly, this pathway is quite distinct from the ROS-Ask1-JNK/p38 axis that regulates Upd3 expression under normal physiological conditions. Our results unravel an unknown physiological role of FAO in cytokine-dependent ECM remodeling, bearing implications in diabetic fibrosis.

## Introduction

The extracellular matrix (ECM) is a dynamic three-dimensional scaffold of fibrous proteins, proteoglycans, and glycoproteins organized in a cell/tissue-specific manner that continually undergoes controlled remodeling. Besides providing physical support, the ECM profoundly influences a diverse array of cellular functions during development and throughout life (Bonnans et al., 2014; Lu et al., 2011; Rozario and DeSimone, 2010). On the flip side, defects in ECM composition cause major developmental defects and strongly contribute to several pathological conditions such as cancer, osteoarthritis, and fibrosis (Bonnans et al., 2014; Frantz et al., 2010; Lu et al., 2011). Understanding the regulatory mechanisms of ECM remodeling is thus essential to comprehend the complex ECM dynamics and its implications in organismal health and disease.

Emerging studies demonstrate that substantial ECM remodeling is associated with increased consumption of high-calorie diets. At times, it could lead to fibrosis, characterized by an excessive or inappropriate accumulation of ECM components in various tissues. For instance, it has been documented that significant changes in cardiac ECM turnover lead to interstitial and perivascular fibrosis due to high dietary sugar (Borghetti et al., 2018; Tate et al., 2017). In both human patients and animal models of diabetes, the roles of diabetes-associated hyperglycemia, advanced glycation end products, reactive oxygen species (ROS), neurohumoral signals, inflammatory cytokines, growth factors, and adipokines have been implicated in the activation of the resident cardiac fibroblasts for the excessive synthesis of cardiac ECM proteins (Russo and Frangogiannis, 2016; Tuleta and Frangogiannis, 2021). Several signaling molecules including TGF β1, Angiotensin II, ROS, and STAT-3 facilitate the process of fibrosis either by enhancing collagen synthesis in the fibroblasts or by inducing their proliferation (Fiaschi et al., 2014; Frangogiannis, 2019, 2021; Russo and Frangogiannis, 2016). Furthermore, other myocardial cell types, including immune cells, cardiomyocytes, endothelial cells, and pericytes, have been documented to contribute to diabetic cardiac fibrosis. Either they are involved in the secretion of profibrotic mediators and fibrogenic growth factors, or they get transdifferentiated into fibroblasts. Nevertheless, the mechanistic basis of high dietary sugar–induced cardiac ECM remodeling and its subsequent contribution to the manifestation of cardiomyopathy remains poorly understood and forms the central focus of this study.

[1]Department of Biological Sciences, Molecular Cell and Developmental Biology Laboratory, Indian Institute of Science Education and Research Mohali, Punjab, India; [2]Department of Biological Sciences, Developmental Genetics Laboratory, Indian Institute of Science Education and Research Mohali, Punjab, India.

*D. Kumar and G. Chauhan contributed equally to this paper.  Correspondence to Sudip Mandal: sudip@iisermohali.ac.in

J. Gera's current affiliation is Neurobiology and Genetics, Theodor-Boveri-Institute, Biocenter, Julius-Maximilians-University of Würzburg, Würzburg, Germany.

The fruit fly, *Drosophila melanogaster*, is a versatile model organism that has been extensively used in biomedical research for the last few decades. In particular, the heart of *Drosophila* provides a simple yet elegant genetic model to examine ECM turnover and fibrosis. The adult heart of a *Drosophila*, also known as the dorsal tube, is a contractile tubular structure that spans the midline of the abdomen, just beneath the dorsal cuticle (Fig. 1 A). Anteriorly, it extends as the aorta into the thorax. On either side, the heart is flanked by a series of pericardial cells (PCs) that primarily function as nephrocytes (Helmstädter et al., 2017; Zhang et al., 2017). A meshwork of proteoglycans, glycoproteins, and fibrous proteins constitute the cardiac ECM surrounding the heart (Pastor-Pareja and Xu, 2011; Rodriguez et al., 1996; Yasothornsrikul et al., 1997) (Fig. 1 B). The cardiac ECM is instrumental in connecting the cardiomyocytes with the adjoining PCs and muscles, restoring diastolic heart diameter, and participating in synchronization of cardiomyocyte contraction (Rotstein and Paululat, 2016). The fibrous protein Pericardin (Prc), with homologies to mammalian collagen IV (Chartier et al., 2002), is a critical component of the cardiac ECM. Loss of Prc affects cardiac morphogenesis and cardiac integrity, leading to abnormal heart function and reduced lifespan (Chartier et al., 2002; Drechsler et al., 2013; Gera et al., 2022). In adult flies, Prc is synthesized and secreted by the fat cells. Quite intriguingly, an interorgan communication network between the PCs and the fat body regulates the expression of *prc* in the fat cells. High levels of physiological ROS in the PCs activate the Ask1-JNK/p38 signaling cascade to trigger the expression of the cytokine Unpaired 3 (Upd3). In turn, Upd3 released by the PCs activates JAK-STAT signaling in the fat cells to turn on the expression of *prc*. Prc released by the fat cells decorates the cardiac ECM (Gera et al., 2022).

Though insect physiology and carbohydrate metabolism bear specific differences with mammals, many studies have evidenced that the molecules and signaling pathways involved in nutrient sensing and sugar homeostasis in *Drosophila* are evolutionarily conserved and functionally similar to those observed in mammals (Chatterjee and Perrimon, 2021; Mattila and Hietakangas, 2017). Analogous to the mammalian insulin/glucagon endocrine system, *Drosophila* insulin-like peptides (dILPs) and glucagon-like adipokinetic hormone play a central role in glucose metabolism (Baker and Thummel, 2007). Moreover, not only are the members of the insulin signaling pathway conserved (Garofalo, 2002), but mutations in the genes coding for them disrupt sugar homeostasis (Böhni et al., 1999; Kim and Neufeld, 2015; Murillo-Maldonado and Riesgo-Escovar, 2017). Strikingly, elevated dietary sugar levels manifest several hallmark features of type 2 diabetes (T2D) and lead to progressive heart failure accompanied by fibrosis-like accumulation of Prc in the cardiac ECM of adult flies (Musselman et al., 2011; Na et al., 2013). However, the molecular and genetic basis of high dietary sugar induced increased Prc accumulation remains elusive.

Our results unravel the mechanistic basis of the process involved in remodeling cardiac ECM upon high dietary sugar intake. Based on in vivo molecular genetic analysis, we demonstrate that high dietary sugar causes ROS-independent activation of JNK signaling to promote elevated levels of fatty acid oxidation (FAO) in the PCs. Importantly, we establish the role of high FAO–induced histone acetylation in transcriptional upregulation of the cytokine Upd3, responsible for enhancing *prc* expression. Thus, besides revealing the molecular genetic basis of high dietary sugar–induced increased accumulation of Prc in the cardiac ECM of *Drosophila*, the outcome of this study sheds light on an otherwise unappreciated physiological role of FAO in ECM remodeling by modulating cytokine levels with far-reaching implications in diabetic fibrosis.

## Results

### High dietary sugar remodels the cardiac ECM due to elevated levels of Prc expression in the fat cells

For our studies, we compared adult flies reared on high sugar diet (HSD; standard cornmeal-sugar-yeast-agar medium brought to a final concentration of 1 M sucrose) to those reared on a normal diet (ND; standard cornmeal-sugar-yeast-agar medium with 0.15 M sucrose). For the first 2 days after eclosion (AE), the control and experimental batches of flies were reared on ND. Subsequently, the experimental flies were transferred to HSD, while the control batches of flies were reared on ND for the rest of their life (Fig. 1 C). The hemolymph glucose level in flies reared on ND ranged between 63.91 ( ±4.8) mg/dl and 96.3 ( ±10.4) mg/dl till day 30AE (Fig. 1 D). In contrast, the flies fed on HSD exhibited a gradual increase in hemolymph glucose level, attaining a more than twofold increase by day 20AE (Fig. 1 D). Additionally, on day 20AE, the flies reared on HSD exhibited hypertrehalosemia (Fig. S1 A), increased triglyceride levels (Fig. S1 B), and insulin resistance (Fig. S1, C and C′) despite an increase in the production of insulin-like peptides, DILP2, DILP3, and DILP5, primarily associated with carbohydrate metabolism (Fig. S1 D). The HSD-fed flies demonstrated shortened average life expectancy (Fig. S1 E), with the median lifespan being 7 days shorter than the ND-fed flies (Fig. S1 F). Importantly, on day 20AE, flies reared on HSD demonstrated an increased amount of Prc accumulation in the cardiac ECM (Fig. 1, E–G) and exhibited irregular beating patterns of the heart (Fig. 1 H and Videos 1 and 2) with a significant increase in arrhythmia index (Fig. 1 I) when compared to the flies reared on ND. Given that by day 20AE, HSD-fed flies demonstrated several hallmark features of T2D, fibrosis-like accumulation of Prc in the cardiac ECM, and cardiac dysfunction, as evidenced earlier (Musselman et al., 2011; Na et al., 2013), for the rest of the study, all analyses were done on day 20AE.

In adult flies, Prc is synthesized and released by the fat cells in a JAK-STAT–dependent manner. The cytokine Upd3, released by the PCs, triggers the JAK-STAT signaling in the fat cells (Fig. 1 J) (Gera et al., 2022). Our investigation revealed that the level of expression of prc is more than threefold higher in the fat cells of HSD-fed flies compared with those reared on ND (Fig. 1 K). In tune with this result, a robust increase in the reporter UAS-GFP expression driven by prc-Gal4 was detected in the fat cells of HSD-fed flies (Fig. 1, L and M). Notably, fat cell–specific attenuation of JAK-STAT signaling by knocking down the expression of either Stat92E (the signal transducer) or Domeless (the receptor) drastically reduced *prc* expression in the

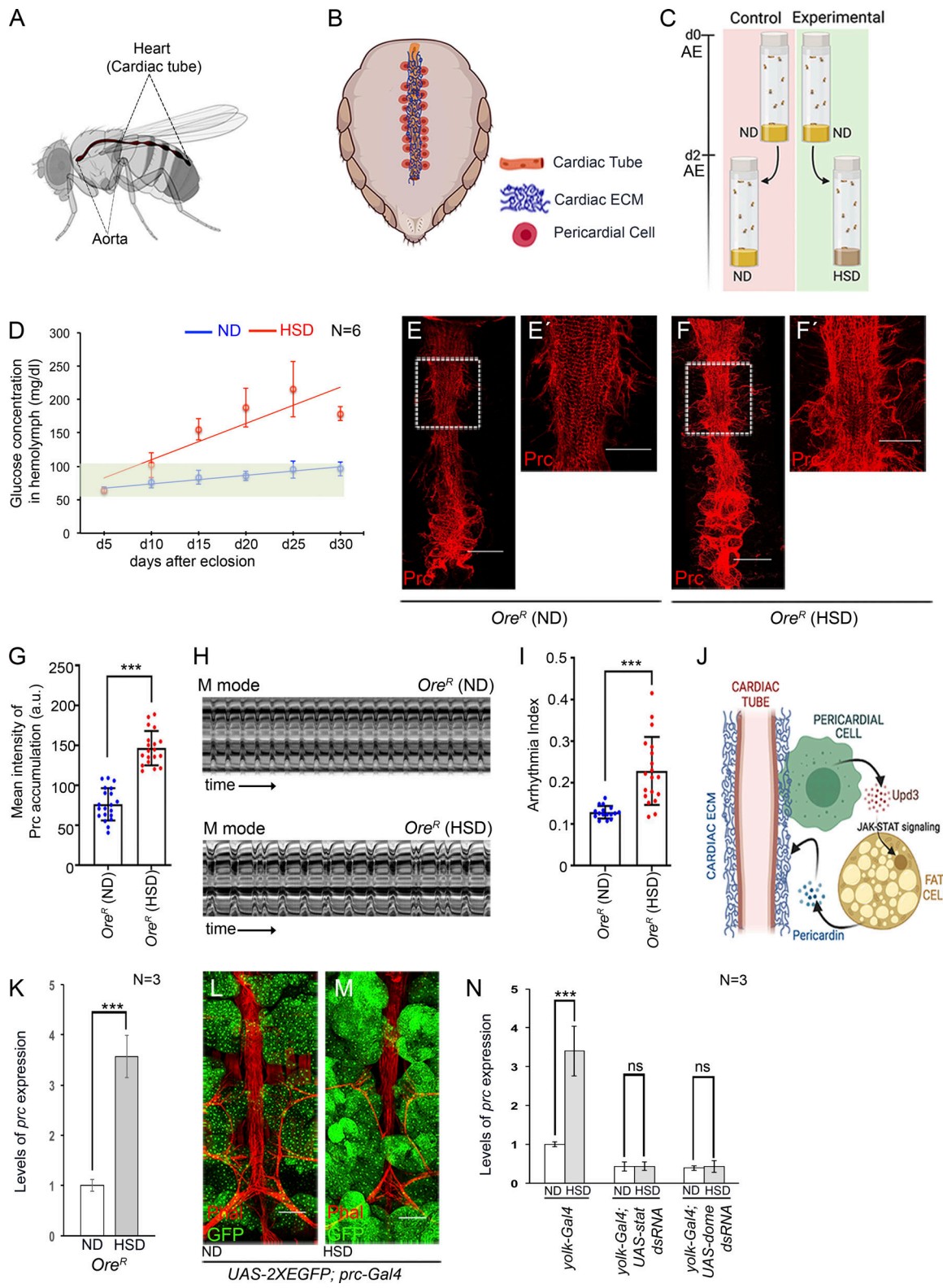

Figure 1. **Increased levels of *prc* expression in fat cells of HSD-fed flies affect cardiac function. (A and B)** Schematic representations showing the anatomical location of the adult *Drosophila* heart (A) and the association of PCs and cardiac ECM with the heart (B). **(C)** Schematic representation of the diet regimen followed to rear flies on high dietary sugar. **(D)** Changes in the hemolymph glucose levels in ND- and HSD-fed flies with age. **(E–F′)** High dietary sugar leads to an increase in Prc (red) accumulation around the heart (F) as compared with that observed in flies reared on ND (E). E′ and F′ represent zoomed-in images of the second cardiac chamber marked with a box in E and F, respectively. Scale bars, 100 μm (E and F) and 50 μm (E′ and F′). **(G)** Quantification of the mean fluorescence intensity for Prc accumulation around the second heart chamber of flies reared on either ND or HSD. The dots represent the samples analyzed for each condition. **(H)** Representative M-mode records for heartbeats of adult flies reared on either ND or HSD, showing the movement of the heart

tube walls (y-axis) over time (x-axis). **(I)** Increase in arrhythmia index of the heart upon rearing the flies on HSD. The dots represent the samples analyzed for each condition. **(J)** Scheme showing the interorgan communication circuitry between the PCs and the fat cells responsible for regulating prc expression from the fat cells under normal physiological conditions. **(K)** Increase in the levels of prc expression in the fat cells of HSD-fed flies. The transcript levels are normalized to that of the constitutive ribosomal gene rp49. **(L and M)** HSD induced an increase in the reporter GFP expression for prc (green) in the fat cells (M) as compared with those reared on ND (L). Phalloidin (red) marks the cardiac tube and the alary muscles. Scale bars, 100 μm. **(N)** Changes in the levels of prc expression upon knocking down either stat92E or dome in the fat cells of flies reared on either ND or HSD. The transcript levels are normalized to that of the constitutive ribosomal gene rp49. Genotypes are as mentioned. Data are represented as mean ± SD. P values (ns ≥ 0.05, ***$P < 1 \times 10^{-3}$) were obtained by unpaired Student's $t$ test (two-tailed) with Welch's correction (G, I, and K) or by two-way ANOVA with Tukey's multiple-comparison test (N). d, day.

fat cells of HSD-fed flies to a level that was comparable with that observed upon similar genetic manipulations in ND-fed flies (Fig. 1 N and Fig. S1 G).

### Elevated levels of Upd3 expression in PCs upregulate *prc* expression in the fat cells of HSD-fed flies

To ascertain whether the upregulation in *prc* expression in the fat cells of HSD-fed flies is induced by increased levels of *upd3* expression in the PCs, we first checked for the levels of *upd3* expression in the PCs. As evident from Fig. 2, A–C, compared with those reared on ND, a marked increase in reporter *lacZ* expression for *upd3* was detected in the PCs of HSD-fed flies. An analogous increase in the transcript levels of *upd3* was detected in the RNA isolated from the PCs and cardiac tube of HSD-fed flies (Fig. 2 D). Considering that *upd3-lacZ* expression is restricted to the PCs, we argue that the enrichment of *upd3* transcripts is primarily due to elevated levels of *upd3* expression only in the PCs. Next, we wanted to downregulate *upd3* specifically in the PCs and check for *prc* expression in the fat cells. For that purpose, we knocked down *upd3* in the PCs of HSD-fed flies by *dot-Gal4*, which specifically expresses in the PCs (Fig. S2 A; Gera et al., 2022). Downregulation of upd3 in the PCs of HSD-fed flies resulted in a drastic reduction in prc expression in the fat cells (Fig. 2 E and Fig. S2 B). Consistent with these results, a remarkable reduction in the excessive amount of Prc accumulation around the cardiac tube of HSD-fed flies was also detected upon knocking down *upd3* in the PCs (Fig. 2, F–I). Together, these results establish that elevated levels of Upd3 from the PCs augment prc expression in a JAK-STAT–dependent manner in the fat cells of HSD-fed flies.

We then sought to determine whether the *upd3* knockdown-dependent suppression of increased *prc* expression could restore normal cardiac function in HSD-fed flies. Though a detectable suppression of the irregularity in the heartbeats of HSD-fed flies was observed upon downregulating pericardial *upd3* expression (Fig. 2 J and Videos 3, 4, and 5), the rate of heartbeats was still slower compared with that observed in the ND-fed flies. While an appreciable recovery was detected in arrhythmia index (Fig. 2 K) and systolic interval (Fig. S2 E), the heart period (Fig. S2 C) and the diastolic interval (Fig. S2 D) were still higher in HSD-fed flies upon knocking down *upd3* in the PCs. We argue that complete restoration in cardiac function was not seen as the observed cardiac dysfunction in HSD-fed flies could be an outcome of several factors. Accumulation of excess Prc in the cardiac ECM might be one of the contributing factors. Nonetheless, these results establish a causal relationship between increased pericardial Upd3 and cardiac dysfunction associated with enhanced Prc accumulation in the cardiac ECM of HSD-fed flies.

### ROS-independent activation of JNK upregulates *upd3* expression in the PCs of HSD-fed flies

Under normal dietary conditions, high physiological levels of ROS trigger a signaling cascade that involves Ask1 and JNK/p38 to regulate *upd3* expression in the PCs (Fig. 3 A). Given that excessive ROS generation and oxidative stress are associated with the pathophysiology of diabetic cardiomyopathy (Kaludercic and Di Lisa, 2020), we conjectured that a further increase in the levels of ROS in the PCs could hyperactivate this signaling cascade to augment *upd3* expression in the PCs of HSD-fed flies. Surprisingly, the intensity of dihydroethidium staining (DHE; a redox-sensitive dye that detects superoxide radicals) was drastically reduced in the PCs of HSD-fed flies (Fig. 3, B–D). This drop in ROS level was further confirmed by analyzing the expression of glutathione S-transferase D1-GFP (*gstD1-GFP*), an in vivo reporter for cellular ROS levels (Sykiotis and Bohmann, 2008). Compared with that observed in the PCs of flies reared on ND, reduced levels of *gstD1-GFP* expression were detected in the PCs of HSD-fed flies (Fig. S2, F–H). On a similar note, a reduction in the level of phospho-p38 was observed in the PCs of HSD-fed flies (Fig. 3, E–G). However, in sharp contrast, a robust increase in TRE-dsRed (reporter for JNK signaling) expression was detected in the PCs of HSD-fed flies (Fig. 3, H–J). Importantly, attenuating JNK signaling by expressing a dominant negative form of *basket* (*bsk*, JNK in flies) resulted in an appreciable restoration in the level of *upd3* expression in the PCs of HSD-fed flies, implicating the involvement of JNK signaling in the upregulation of *upd3* expression (Fig. 3 K and Fig. S2 I). Analogous results were obtained when the expression of *kayak* (*kay*, Fos in flies) was downregulated in the PCs of HSD-fed flies (Fig. 3 K and Fig. S2 I).

Apoptotic signaling kinase 1 (Ask1) is a Jun N-terminal Kinase Kinase that serves as a ROS sensor and gets activated via ROS-mediated dimerization (Sekine et al., 2012). Inactivating Ask1 by knocking down its expression did not lead to any drop in the level of TRE-dsRed expression in the PCs of HSD-fed flies (Fig. 3, L and O). Furthermore, attenuating Ask1 function in the PCs of HSD-fed flies failed to rescue the elevated levels of *upd3* transcripts observed in the RNA isolated from the PCS and cardiac tube (Fig. 3 P and Fig. S2 J). Together, these results demonstrate that the upregulation of *upd3* expression in the PCs of HSD-fed flies is not mediated by the conventional ROS-Ask1-JNK/p38 signaling cascade. Instead, elevated JNK signaling, triggered in a ROS-independent manner, induces the *upd3* expression to potentiate the excessive expression of Prc. Importantly, this kind of regulation appears quite distinct from the previously known roles of ROS and oxidative stress as common denominators associated with cardiac hypertrophy and cardiac fibrosis (Wilson et al., 2018; Zhou et al., 2011).

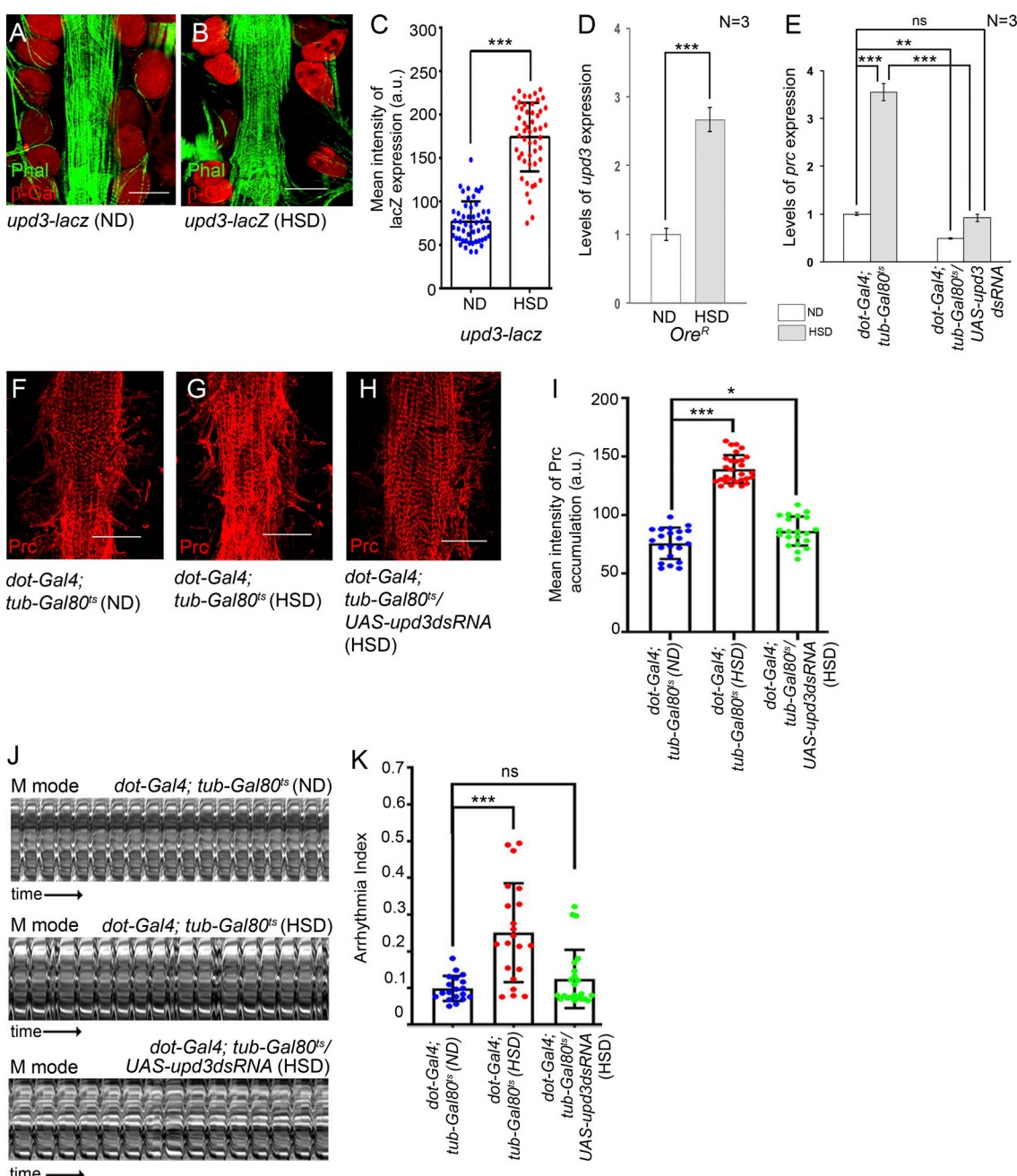

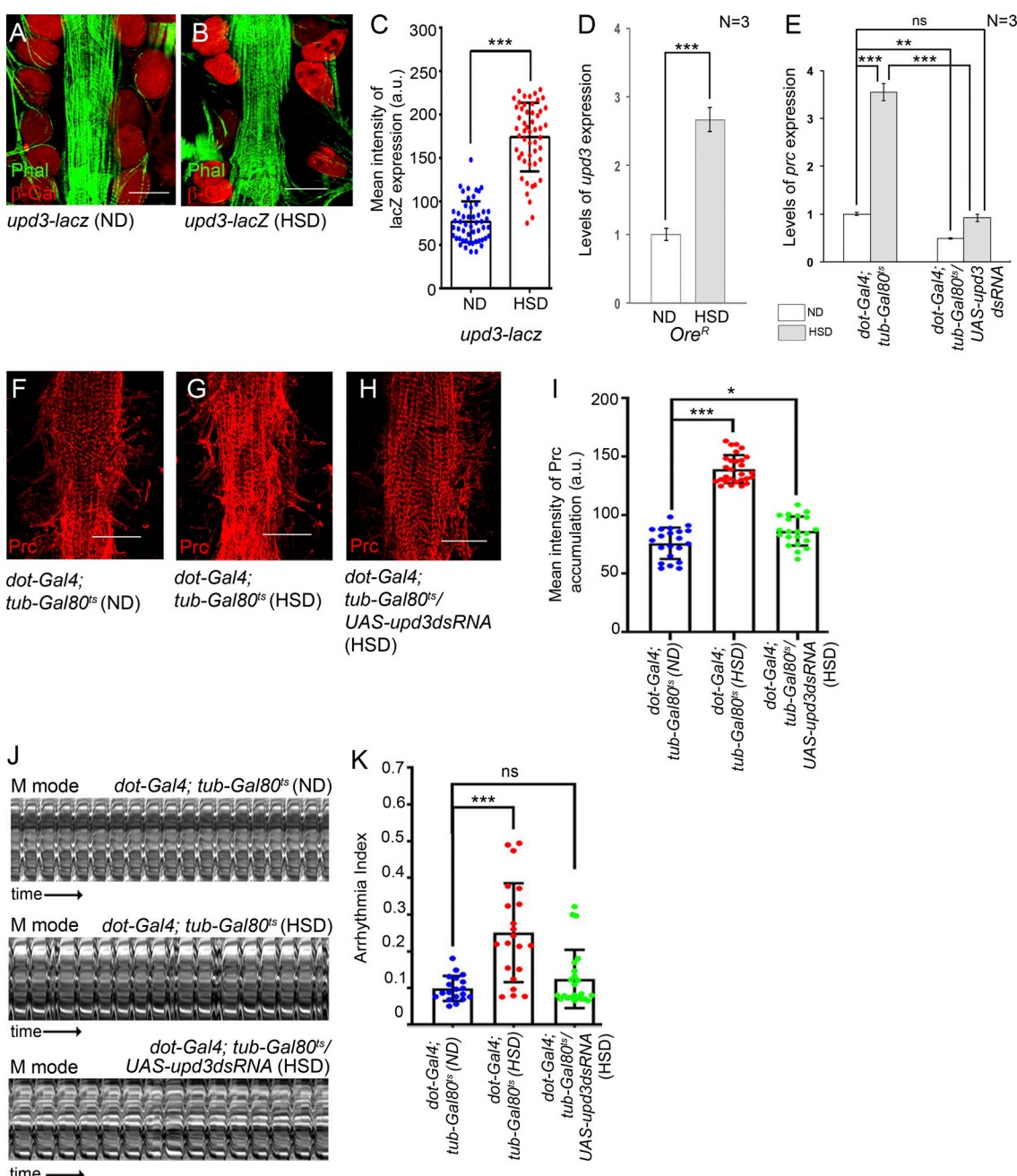

Figure 2. **Increased levels of pericardial Upd3 upregulate *prc* expression in fat cells of HSD-fed flies. (A and B)** Increase in the reporter lacZ expression for *upd3* (red) in the PCs of HSD-fed flies (B) as compared with flies reared on ND (A). Phalloidin (green) marks the heart and the alary muscles. Scale bars, 50 µm. **(C)** Quantification of the mean fluorescence intensity for *upd3-lacZ* expression in the PCs of flies reared on either ND or HSD. The dots represent the number of PCs quantified for each condition. **(D)** Increase in the levels of *upd3* expression in the PCs of HSD-fed flies as compared with that observed upon rearing on ND. The transcript levels are normalized to that of the constitutive ribosomal gene *rp49*. **(E)** Changes in the levels of *prc* expression in the fat cells upon knocking down *upd3* in the PCs of flies reared on either ND or HSD. The transcript levels are normalized to that of the constitutive ribosomal gene *rp49*. **(F–H)** Changes in Prc (red) accumulation around the heart in HSD-fed flies upon knocking down *upd3* in the PCs (H) as compared with that observed in HSD-fed flies (G), and upon rearing them on ND (F). Scale bars, 50 µm. **(I)** Quantification of the mean fluorescence intensity for Prc accumulation around the second heart chamber in HSD-fed flies upon knocking down *upd3* in the PCs. The dots represent the samples analyzed for each genotype. **(J)** Representative M-modes for heartbeats of adult flies showing the movement of the heart tube walls (y-axis) over time (x-axis). **(K)** Changes in arrhythmia index in the hearts of HSD-fed flies upon knocking down *upd3* in the PCs. The dots represent the samples analyzed for each genotype. Genotypes are as mentioned. Data are represented as mean ± SD. P values (ns ≥ 0.05, *P < 0.05, **P < 0.01, ***P < 1 × 10⁻³) were obtained by unpaired Student's *t* test (two-tailed) with Welch's correction (C and D) or by two-way ANOVA with Tukey's multiple-comparison test (E) and one-way ANOVA with Tukey's multiple-comparison test (I and K).

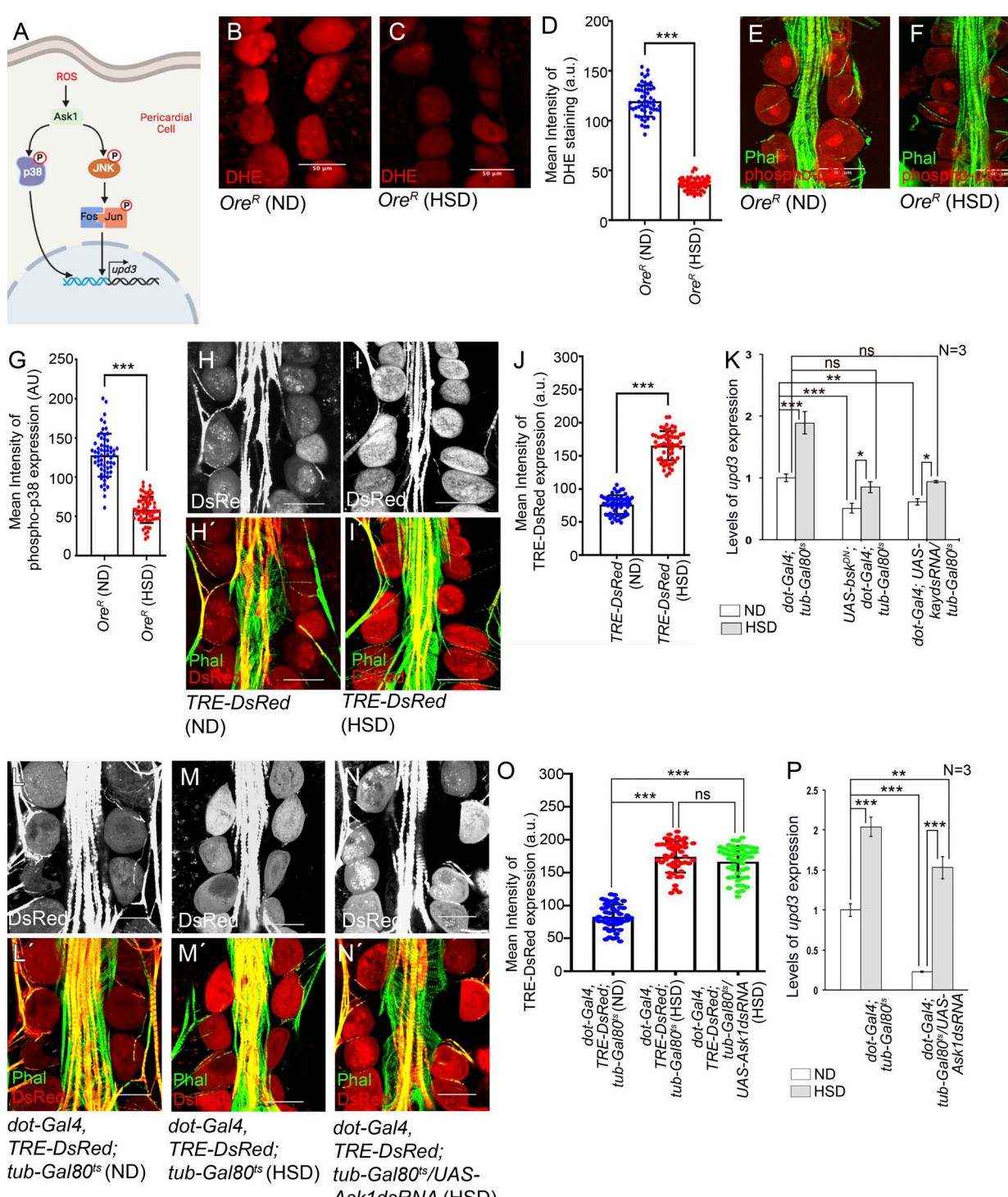

Figure 3. **ROS-independent activation of JNK signaling triggers pericardial *upd3* expression. (A)** Diagram showing the signaling cascade that involves ROS, Ask1, and JNK/p38 in activating *upd3* expression in the adult PCs during normal physiological conditions. **(B and C)** Drop in the levels of ROS (DHE staining; red) in the PCs of HSD-fed flies (C) as compared with those reared on ND (B). **(D)** Quantification of the mean fluorescence intensity for DHE staining in the PCs of flies reared on either ND or HSD. The dots represent the number of PCs analyzed for each genotype. **(E and F)** Drop in the levels of phosphorylated p38 (red) in the PCs of HSD-fed flies (F) as compared to those reared on ND (E). Phalloidin (green) marks the cardiac tube and the alary muscles. **(G)** Quantification of the mean fluorescence intensity for phosphorylated p38 levels in the PCs of flies reared on either ND or HSD. The dots represent the number of PCs analyzed for each genotype. **(H–I′)** Increase in the expression of TRE-dsRed (red; reporter for JNK pathway) in the PCs of HSD-fed flies (I and I′) as compared to those reared on ND (H and H′). Phalloidin (green) marks the cardiac tube and the alary muscles. H and I show TRE-dsRed expression in grayscale; H′ and I′ represent the merged images of the two colors. **(J)** Quantification of the mean fluorescence intensity for TRE-DsRed expression in the PCs of flies reared on either ND or HSD. The dots represent the number of PCs analyzed for each genotype. **(K)** Changes in the level of upd3 expression in the PCs of flies reared on ND or HSD upon attenuating JNK signaling. **(L–N′)** Changes in TRE-dsRed expression (red) in the PCs of HSD-fed flies upon PC-specific

attenuation of Ask1 (N and N′) as compared with that observed in HSD-fed flies (M and M′), and upon rearing them on ND (L and L′). Phalloidin (green) marks the cardiac tube and the alary muscles. L, M, and N show TRE-dsRed expression in grayscale; L′, M′, and N′ represent the merged images of the two colors. **(O)** Quantification of the mean fluorescence intensity for TRE-DsRed expression in the PCs of HSD-fed flies upon attenuating Ask1. The dots represent the number of PCs analyzed for each genotype. **(P)** Changes in the level of upd3 expression in the PCs of flies reared on ND or HSD upon PC-specific attenuation of Ask1. Genotypes are as mentioned. Data are represented as mean ± SD. P values (ns ≥ 0.05, *P < 0.05, **P < 0.01, ***P < 1 × 10⁻³) were obtained by unpaired Student's $t$ test (two-tailed) with Welch's correction (D, G, and J) or by two-way ANOVA with Tukey's multiple-comparison test (K, O, and P). Scale bars, 50 µm.

## Pericardial cells of HSD-fed flies exhibit compromised glycolysis

Our results demonstrated that feeding adult flies on HSD led to insulin resistance, a physiological condition characterized by reduced tissue responsiveness to insulin and low glucose uptake. To analyze the glucose uptake potential of the PCs in HSD-fed flies, we stained them with 2-NBDG, a fluorescent deoxyglucose analog (Zou et al., 2005) that can be taken up by cells through glucose transporters. The PCs of HSD-fed flies demonstrated a threefold reduction in the uptake of 2-NBDG compared with those of flies reared on ND (Fig. 4, A–C) as reported earlier (Na et al., 2015). Next, we used a GLUT1-GFP protein trap reporter to analyze the expression of glucose transporters. As evident from Fig. 4, D–F, compared with the PCs of ND-fed flies, the PCs of HSD-fed flies exhibited a significant drop in GLUT1-GFP expression. Similarly, the PCs of HSD-fed flies exhibited a remarkable drop in the expression of the Hexokinase-GFP (HexA-GFP; Hexokinase is the initial glycolytic enzyme involved in catalyzing the phosphorylation of glucose to glucose-6-phosphate) as compared with that observed in the PCs of ND-fed flies (Fig. 4, G–I). An analogous drop in reporter GFP expression for Phosphoglucoisomerase (Pgi), the glycolytic enzyme responsible for the reversible isomerization of glucose-6-phosphate and fructose-6-phosphate, was also detected in the PCs of HSD-fed flies (Fig. 4, J–L) Together, these results indicate that the process of glycolysis is compromised in the PCs of the HSD-fed flies.

## PCs of HSD-fed flies exhibit altered lipid metabolism

Several studies demonstrated that besides reduced glucose uptake, insulin resistance and hyperglycemic conditions also alter cellular lipid metabolism, leading to the development of dyslipidemia in mammals (Schofield et al., 2016). Given that the basic metabolic pathways and signaling pathways involved in lipid metabolism are evolutionarily and functionally conserved between *Drosophila* and mammals (Chatterjee and Perrimon, 2021), we were intrigued to investigate the status of lipid metabolism in the PCs of HSD-fed flies. Cellular lipid metabolism in *Drosophila* is divided into two main processes: one is the formation of triglycerides (TAG) (lipid droplets) called lipogenesis and the other is lipolysis or mobilization of TAG from lipid droplet (Fig. 5 A). Quantitative analyses of the transcript levels of the genes encoding enzymes involved in lipogenesis, which include *Glycerol-3-phosphate acyltransferase 4* (Gpat4), *Acylglycerol phosphate acyltransferase 4* (Agpat4), and *Diacylglycerol acyltransferase* (Dgat2) (Fig. 5 A), revealed their enrichment in the RNA isolated from PCs and cardiac tube of HSD-fed flies as compared with that of ND-fed flies (Fig. 5 B). Interestingly, compared with Gpat4 and Agpat4, a robust increase in the levels of Dgat2 transcripts (which codes for the enzyme involved in the

conversion of diacylglycerol (DAG) to triacylglycerol was detected (Fig. 5 B). Interestingly, expression of the genes involved in lipolysis, such as *Brummer lipase* (bmm) and *Hormone-sensitive lipase* (Hsl) (Fig. 5 A), were also upregulated in HSD-fed condition (Fig. 5 B). In contrast, a significant reduction in the levels of *Acetyl-Coenzyme A carboxylase* (Acc) and *Fatty acid synthase 1* expression (genes involved in de novo fatty acid synthesis; Fig. 5 A) was observed (Fig. 5 B). Together these results indicate high dietary sugar induced significant alterations in cellular lipid metabolism; while the kinetics of both lipogenesis and lipolysis are upregulated, generation of fatty acids by de novo fatty acid synthesis is compromised. Since the rate of de novo fatty acid synthesis is dependent on glycolytic flux (Fig. 5 A), low sugar uptake and a consequent impairment in glycolysis might be responsible for the reduction in de novo fatty acid synthesis.

To substantiate the results of these expression studies, we first stained these cells with Boron-Dipyrromethane (BODIPY), a selective fluorescent stain for intracellular neutral lipid droplets. While low staining levels were observed in the PCs of flies reared on ND (Fig. 5, C–E), a robust increase in neutral lipids was detected in the PCs of HSD-fed flies (Fig. 5, D and E). The lipid enrichment in the PCs of HSD-fed flies was further evident upon LipidTOX (a validated marker for neutral lipids) staining (Fig. S3, A–C). A robust increase in the triacylglycerol level was also evidenced in the tissue that included the cardiac tube and the PCs (Fig. 5 F). Next, we checked for the levels of free fatty acids by staining the tissues with Nile Blue A (Boumelhem et al., 2022). As evident from Fig. 5, G and I, a significant increase in Nile Blue A staining was observed in the PCs of HSD-fed flies as compared with that observed in ND-fed flies. Therefore, these results, in conjunction with the upregulation in the expression of genes associated with lipogenesis and lipolysis, demonstrate that lipid metabolism gets altered in the PCs of the HSD-fed flies.

Cells, when experiencing low glycolytic flux, generally draw upon lipids as an alternate energy source. Lipid metabolism entails the conversion of stored TAG into fatty acids, followed by FAO in the mitochondria. Acetyl-CoA, one of the end products of FAO, feeds into the Krebs cycle to support energy generation by oxidative phosphorylation. To ascertain whether the PCs of HSD-fed flies rely on FAO, we checked for the reporter GFP expression in the PCs of *Hnf4-Gal4, UAS-mCD8GFP* flies reared on HSD. Compared with the flies reared on ND, elevated levels of reporter GFP expression were observed in the PCs of HSD-fed flies (Fig. 5, J–L). The gene *Hepatocyte nuclear factor 4* (Hnf4) codes for a highly conserved nuclear receptor that regulates lipid mobilization and FAO (Palanker et al., 2009). In flies, the gene *scully* codes for the homolog of mammalian Hydroxy acyl-CoA dehydrogenase, an essential enzyme involved in FAO (Fig. 5 M). Feeding on HSD led to an almost twofold increase in the scully-YFP

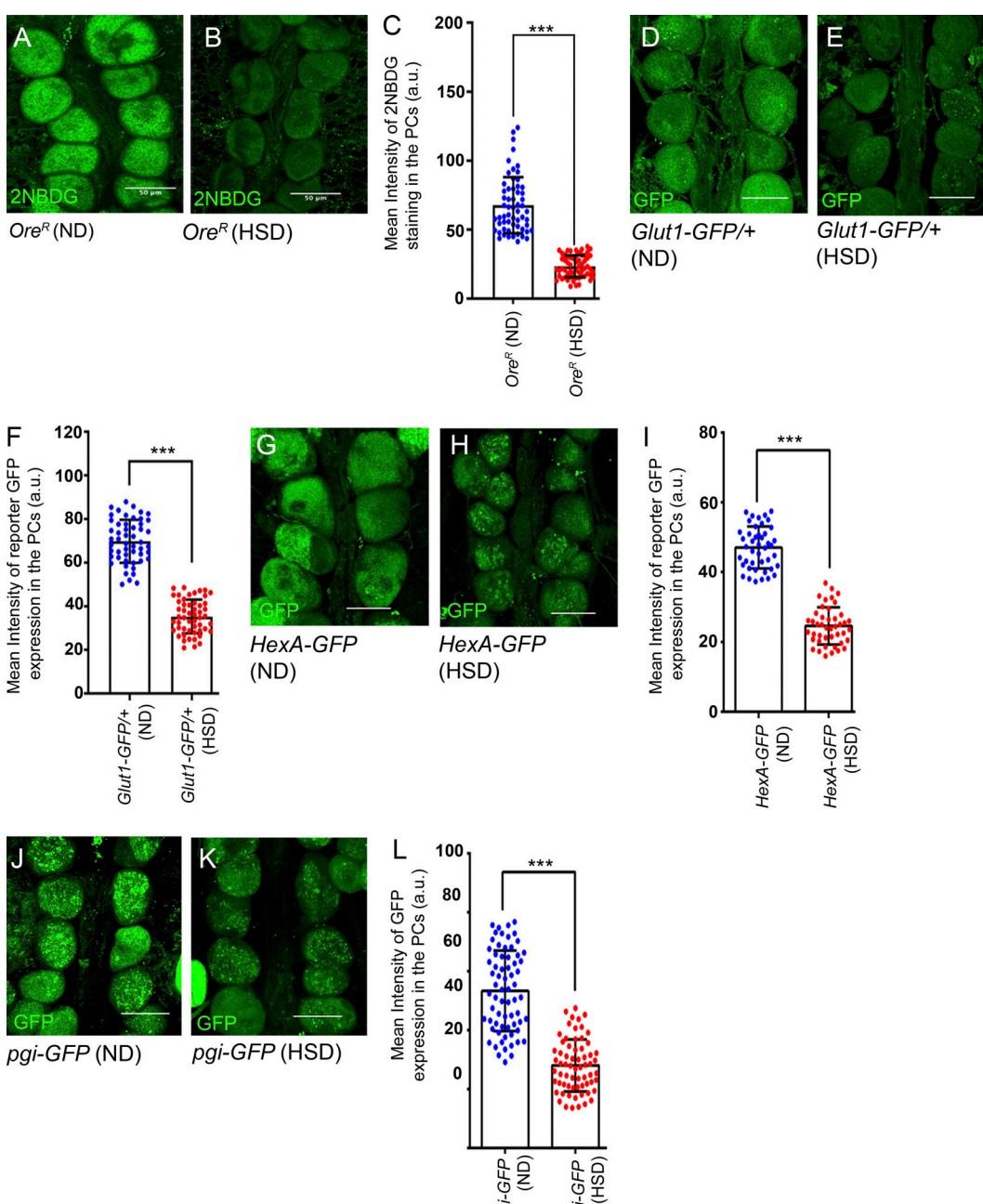

**Figure 4.** **The PCs of HSD-fed flies exhibit low sugar uptake and compromised glycolysis. (A and B)** Drop in the levels of glucose uptake (2-NBDG staining; green) in the PCs of HSD-fed flies (B) as compared to those reared on ND (A). **(C)** Quantification of the mean fluorescence intensity for 2-NBDG in the PCs of flies reared on either ND or HSD. The dots represent the number of PCs analyzed for each genotype. **(D and E)** Drop in the reporter GFP expression (green) for Glut1 in the PCs of HSD-fed flies (E) as compared with those reared on ND (D). **(F)** Quantification of the mean fluorescence intensity for Glut1-GFP expression in the PCs of flies reared on either ND or HSD. The dots represent the number of PCs analyzed for each genotype. **(G and H)** Drop in the reporter GFP expression for HexA in the PCs of HSD-fed flies (H) as compared with those reared on ND (G). **(I)** Quantification of the mean fluorescence intensity for HexA-GFP expression in the PCs of flies reared on either ND or HSD. The dots represent the number of PCs analyzed for each genotype. **(J and K)** Drop in the reporter GFP expression for *pgi* in the PCs of HSD-fed flies (K) as compared with those reared on ND (J). **(L)** Quantification of the mean fluorescence intensity for *pgi-GFP* expression in the PCs of flies reared on either ND or HSD. The dots represent the number of PCs analyzed for each genotype. Genotypes are as mentioned. Data are represented as mean ± SD. P values (***P < 1 × 10$^{-3}$) were obtained by unpaired Student's *t* test (two-tailed) with Welch's correction (C, F, I, and L). Scale bars, 50 μm.

protein trap reporter expression in the PCs as compared with that observed in the PCs of flies reared on ND (Fig. 5, N and P). Similar enrichment in the levels of reporter YFP expression for acyl-CoA dehydrogenase (CG3902) was also observed in the PCs of HSD-fed flies (Fig. S3, D–F). Furthermore, compared with that observed for ND-fed flies, elevated levels of expression of the critical genes involved in FAO (Fig. 5 M), which include *withered* (*whd*), *medium-chain acyl-CoA dehydrogenase, mitochondrial trifunctional protein α*

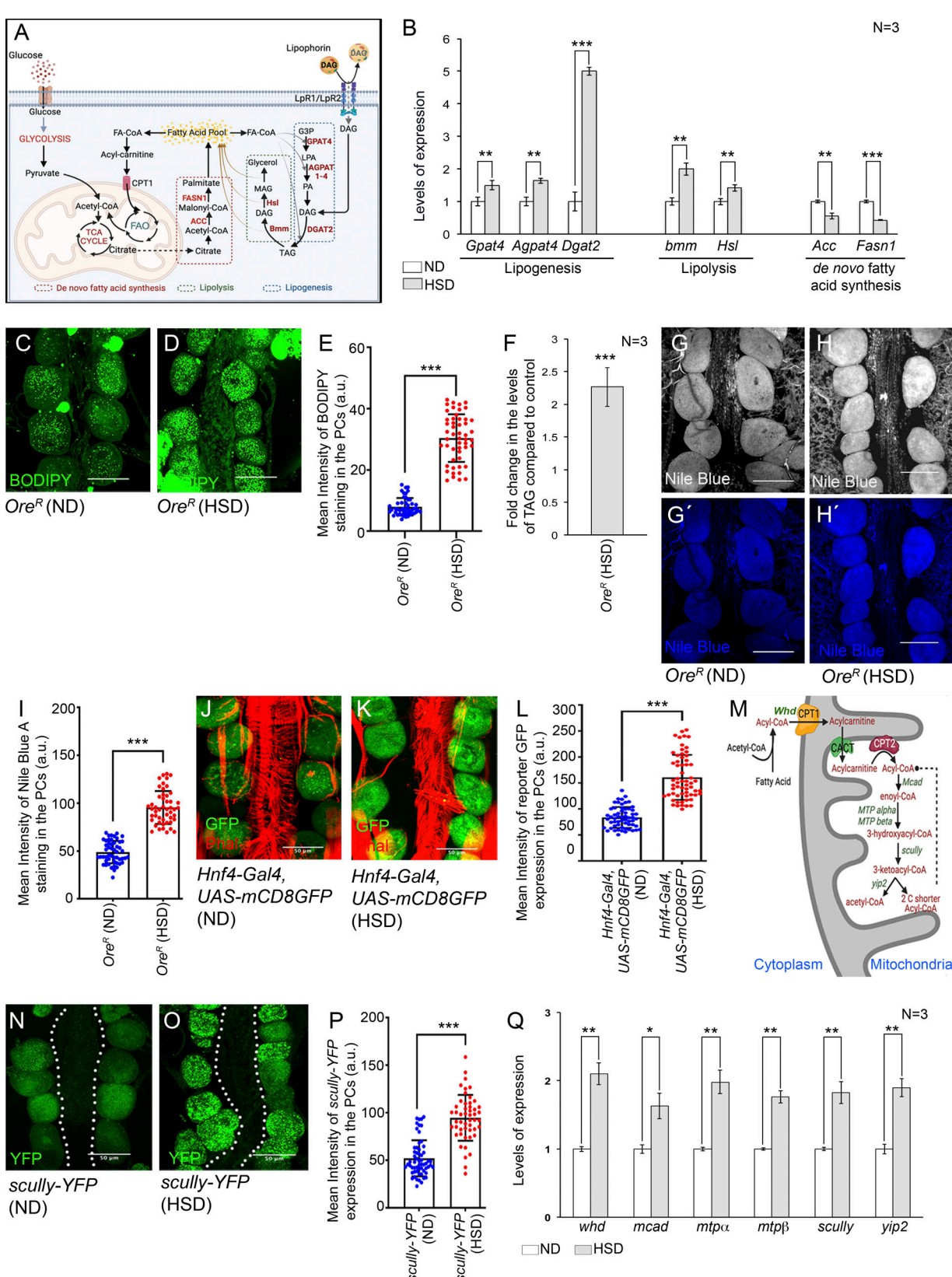

**Figure 5. PCs of HSD-fed flies exhibit altered lipid metabolism and increased FAO. (A)** Schematic representation demonstrating the relation between major processes involved in lipid and carbohydrate metabolism in *Drosophila* cells. For lipid metabolism, these processes include uptake of neutral lipids from the lypophorins by lipophorin receptors (LpR1/LpR2), lipogenesis, lipolysis, de novo fatty acid synthesis, and mobilization of fatty acids (FA) into mitochondria for FAO to feed into the TCA cycle. The enzymes involved in these processes are marked in red. Import of glucose through the glucose transporter, followed by

glycolysis leads to the production of pyruvate that enters the mitochondria to support TCA cycle. **(B)** Changes in the levels of transcripts for the genes involved in lipogenesis, lipolysis, and de novo fatty acid synthesis in the RNA isolated from PCs and cardiac tubes of HSD-fed flies as compared with that in ND-fed flies. **(C and D)** Increase in the levels of lipid accumulation (BODIPY staining; green) in the PCs of HSD-fed flies (D) as compared with those reared on ND (C). **(E)** Quantification of the mean fluorescence intensity for BODIPY staining in the PCs flies reared on either ND or HSD. The dots represent the number of PCs analyzed for each genotype. **(F)** Increase in TAG levels in the PCs and cardiac tube of HSD-fed flies as compared to controls. **(G–H′)** Increase in the levels of free fatty acid (blue; Nile Blue A staining) in the PCs of HSD-fed flies (H and H′) as compared to those reared on ND (G and G′). G and H show Nile Blue A expression in grayscale; G′ and H′ represent the same images in blue channel. **(I)** Quantification of the mean fluorescence intensity for Nile Blue A staining in the PCs flies reared on either ND or HSD. The dots represent the number of PCs analyzed for each genotype. **(J and K)** Increase in the reporter GFP expression (green) for *Hnf*-Gal4 in the PCs of HSD-fed flies (K) as compared with those reared on ND (J). Phalloidin (red) marks the cardiac tube and the alary muscles. **(L)** Quantification of the mean fluorescence intensity for *Hnf*-Gal4,*UAS-mCD8GFP* expression in the PCs of ND or HSD-fed flies. The dots represent the number of PCs analyzed for each genotype. **(M)** Schematic diagram showing the different enzymes involved in the FAO pathway. **(N and O)** Increase in the reporter YFP expression for *scully* in the PCs of HSD-fed flies (O) as compared to those fed on ND (N). **(P)** Quantification of the mean fluorescence intensity for *scully*-YFP expression in the PCs of flies reared on either ND or HSD. The dots represent the number of PCs analyzed for each genotype. **(Q)** Increase in the transcript levels of the genes encoding different enzymes involved in FAO in the RNA isolated from PCs and cardiac tube of flies reared on HSD as compared with those reared on ND. The transcript levels are normalized to that of the constitutive ribosomal gene *rp49*. Genotypes are as mentioned. Data are represented as mean ± SD. P values (*P < 0.05, **P < 0.01, ***P < 1 × 10$^{-3}$) were obtained by unpaired Student's *t* test (two-tailed) with Welch's correction (B, E, F, I, L, P, and Q). Scale bars, 50 μm.

subunit (*Mtpα*), mitochondrial trifunctional protein β subunit (*Mtpβ*), *scully*, and *yippee interacting protein 2*, were detected in the RNA isolated from cardiac tissue and the PCs of HSD-fed flies (Fig. 5 Q). Based on these results, we conclude that the PCs of HSD-fed flies exhibit altered lipid metabolism and rely on elevated levels of FAO for their energy demands.

**Altering lipid uptake and increasing lipolysis impacts high levels of *upd3* expression in the PCs of HSD-fed flies**

In *Drosophila*, lipids are transported in hemolymph as lipoprotein particles, the most abundant being lipophorin. About 95% of all hemolymph lipids are carried by the lipophorin particles predominantly in the form of DAG between organs depending on the changing physiological requirements (Palm et al., 2012; Rodríguez-Vázquez et al., 2015). Uptake of neutral lipids from the circulating lypophorins is mediated by the Lipophorin receptors, LpR1 and LpR2 (members of the low-density lipoprotein receptor family) present on cell surfaces (Fig. 5 A) by an endocytosis-independent mechanism yet to be fully characterized (Parra-Peralbo and Culi, 2011). In an attempt to modulate lipid metabolism in the PCs of HSD-fed flies, we independently knocked down the expressions of *LpR1* and *LpR2*. Downregulation of either of these two genes resulted in a detectable drop in the levels of neutral lipids in the PCs of HSD-fed flies as evidenced by staining with BODIPY (Fig. 6, A–E). In tune with these results, a reduction in the levels of free fatty acids was detected in the PCs of HSD-fed flies upon independent downregulation of *LpR1* and *LpR2* (Fig. 6, F–J). In this context, it is important to note that downregulation of either of these genes failed to completely restore the levels of neutral lipids and free fatty acids to that observed in the PCs of flies reared on ND as these two genes are partially redundant (Parra-Peralbo and Culi, 2011). Next, we wanted to determine whether impairing lipid import into the PCs has any impact on the expression of *upd3*. Indeed, the increased levels of *upd3* transcripts observed in the PCs of HSD-fed flies were partially suppressed upon downregulation of either *LpR1* or *LpR 2* (Fig. 6 K and Fig. S3 G). From these results, we conclude that impairing lipid uptake in the PCs of HSD-fed flies impedes the formation of lipid droplets and free fatty acids, eventually leading to partial restoration of the increased levels of *upd3* expression.

In a converse set of experiments, we overexpressed *bmm* in the PCs of HSD-fed flies in an attempt to increase the levels of free fatty acids by enhanced lipolysis. As evident from Fig. 6, L–N, overexpression of *bmm* led to a remarkable reduction in the levels of neutral lipids in the PCs of HSD-fed flies. The amount of neutral lipids present in the PCs of these flies was even lower than that found in the PCs of flies reared on ND (Fig. 6 O). Concurrent with this result, a significant increase in the levels of free fatty acids was observed in the PCs of the HSD-fed flies (Fig. 6, P–R). More importantly, the elevated levels of *upd3* expression associated with the PCs of HSD-fed flies got further augmented due to overexpression of *bmm* (Fig. 6 S and Fig. S3 H).

In sum, the results of these experiments provide a better understanding of the altered metabolic status in the PCs of HSD-fed flies and how it influences upd3 expression in these cells. The development of insulin resistivity due to high dietary sugar impedes sugar uptake by the PCs, resulting in a consequent drop in glycolytic flux. Under this situation, the PCs demonstrate increased uptake of neutral lipids from circulating lipophorins and convert them into TAGS by lipogenesis. Since the lipophorin particles carry the lipids predominantly in the form of DAG, compared with other genes involved in lipogenesis, we see a robust increase in the expression of Dgat2 responsible for converting DAG to TAG. BODIPY and LipidTOX stainings further endorse the presence of a large number of lipid droplets in these cells. Importantly, these PCs also demonstrate an enhanced rate of TAG mobilization into free fatty acids (FFA). Apart from an increase in the transcript levels of the lipolytic genes (bmm, Hsl), high levels of FFA are detected in these cells. Eventually, the increased levels of FFA fuel FAO to generate energy. Most intriguingly, the elevated levels of FFA upregulate upd3 expression in the PCs of the HSD-fed flies. Therefore, these results demonstrate that high dietary sugar induced increased lipid uptake and its mobilization to FFA in the PCs contribute to upregulating *upd3* expression.

**Elevated levels of *upd3* expression are dependent on FAO in the PCs of HSD-fed flies**

Next, we investigated whether increased FAO, in any way, is responsible for the upregulated *upd3* expression in the PCs of HSD-fed flies. Knocking down the expression of *whd*, which

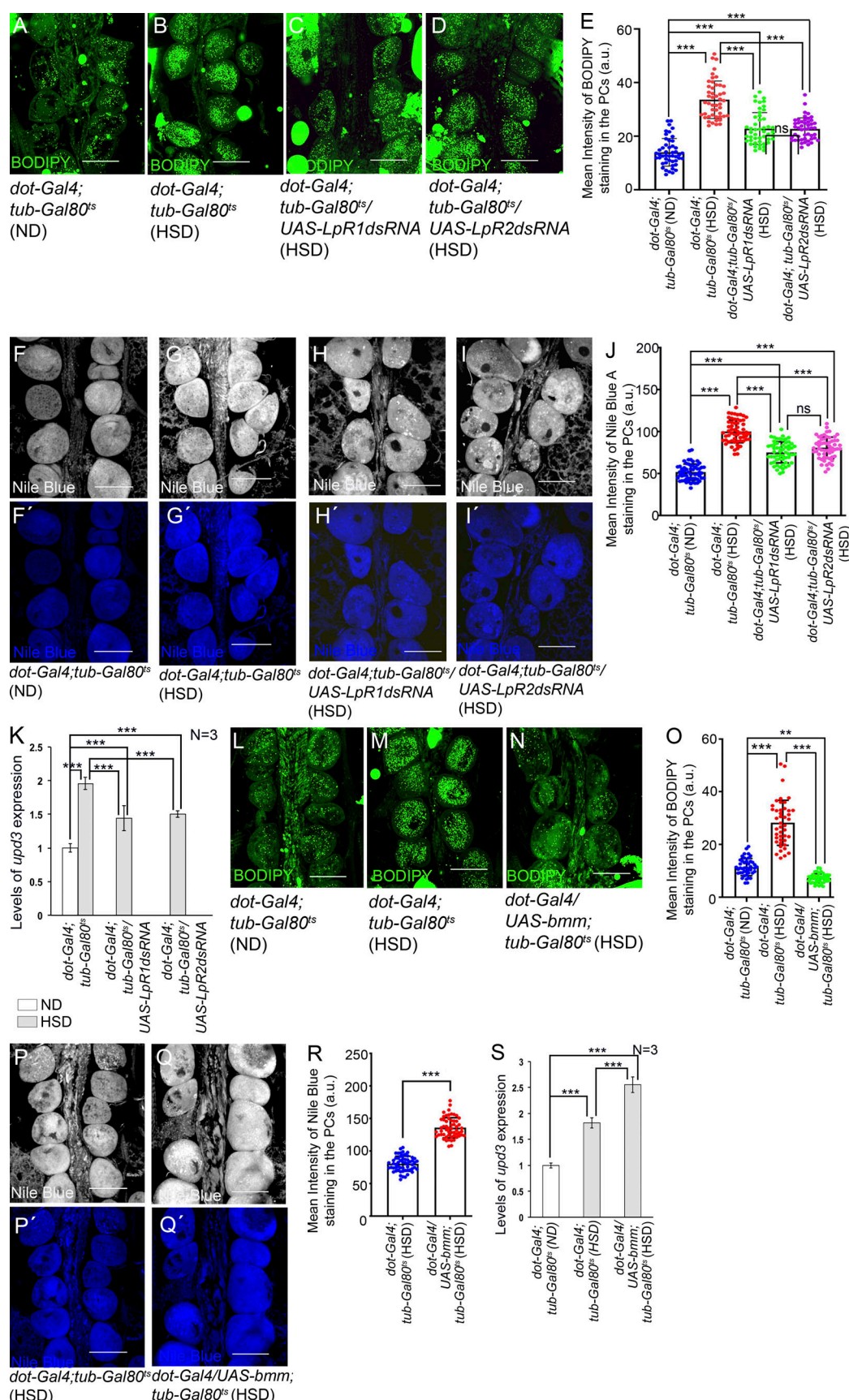

Figure 6. **Increased levels of *upd3* expression in the PCs of HSD-fed flies is dependent on lipid uptake and lipolysis. (A–D)** Changes in the levels of neutral lipid accumulation (green; BODIPY staining) in the PCs of HSD-fed flies (B) and upon downregulation of either *LpR1* (C) or *LpR2* (D) in the PCs of HSD-fed

flies as compared to that observed in ND-fed flies (A). **(E)** Quantification of the mean fluorescence intensity for BODIPY staining in the PCs of flies reared on ND, HSD, and upon downregulation of *LpR1* or *LpR2* in the PCs of HSD-fed flies. The dots represent the number of PCs analyzed for each genotype. **(F–I')** Changes in the levels of Nile Blue A staining in the PCs of HSD-fed flies (G and G') and upon downregulation of either *LpR1* (H and H') or *LpR2* (I and I') in the PCs of HSD-fed flies as compared to that observed in ND-fed flies (F and F'). F, G, H, and I Show Nile Blue A expression in grayscale; F', G', H', and I' represent the same images in blue channel. **(J)** Quantification of the mean fluorescence intensity for Nile Blue A staining in the PCs of flies reared on ND, HSD, and upon downregulation of either *LpR1* or *LpR2* in the PCs of HSD-fed flies. The dots represent the number of PCs analyzed for each genotype. **(K)** Changes in the levels of *upd3* expression in the PCs of flies reared on HSD, and upon downregulating either *LpR1* or *LpR2* in the PCs of HSD-fed flies as compared to that observed in flies reared on ND. The transcript levels are normalized to that of the constitutive ribosomal gene *rp49*. **(L–N)** Decrease in the levels of neutral lipid accumulation (BODIPY staining; green) in the PCs of HSD-fed flies upon overexpression of *bmm* (N) as compared to that observed in HSD-fed flies (M) and in ND-fed flies (L). **(O)** Quantification of the mean fluorescence intensity for BODIPY staining in the PCs of flies reared on ND, HSD, and upon overexpression of *bmm* in the PCs of HSD-fed flies. The dots represent the number of PCs analyzed for each genotype. **(P–Q')** Increase in the levels of Nile Blue A staining in the PCs of HSD-fed flies upon overexpression of *bmm* (Q and Q') as compared to that observed in HSD-fed flies (P and P'). P and Q show Nile Blue A expression in grayscale; P' and Q' represent the same images in blue channel. **(R)** Quantification of the mean fluorescence intensity for Nile Blue A staining in the PCs of flies reared on HSD, and upon overexpression of *bmm* in the PCs of HSD-fed flies. The dots represent the number of PCs analyzed for each genotype. **(S)** Increase in the levels of *upd3* expression in the PCs of flies reared on HSD, and upon overexpression of *bmm* in the PCs of HSD-fed flies as compared to that observed in flies reared on ND. The transcript levels are normalized to that of the constitutive ribosomal gene *rp49*. Genotypes are as mentioned. Data are represented as mean ± SD. P values (ns ≥ 0.05, **P < 0.01, ***P < 1 × 10$^{-3}$) were obtained by unpaired Student's *t* test (two-tailed) with Welch's correction (R) or by two-way ANOVA with Tukey's multiple-comparison test (E, J, K, O, and S). Scale bars, 50 μm.

codes for a rate-limiting enzyme for FAO (Strub et al., 2008), specifically in the PCs of HSD-fed flies, resulted in a robust drop in the levels of *upd3* transcripts in the RNA isolated from PCs and cardiac tube (Fig. 7 A and Fig. S4 A). However, knocking down *whd* in the PCs of flies reared on ND had no impact on the expression of *upd3* (Fig. 7 A). Similarly, even though no change in *upd3* expression was detected upon CRISPR/Cas9-mediated knockout (KO) of *whd* in the PCs of ND-fed flies, a substantial reduction in *upd3* expression was detected when *whd* was knocked out in the PCs of HSD-fed flies (Fig. 7 A and Fig. S4 A). Additionally, *upd3* expression was also analyzed in the RNA isolated from PCs and cardiac tube of *whd1* homozygous null mutant flies. While the expression of *upd3* in *whd1* mutants reared on ND was comparable with that of wild-type control, loss of *whd* caused a significant reduction in the levels of *upd3* expression in the PCs of HSD-fed flies (Fig. 7 B). To further substantiate these observations, we analyzed *upd3* expression in the mutants of two essential enzymes of FAO: *Mtpα* and *Mtpβ* (Kishita et al., 2012). The levels of *upd3* transcripts in the RNA isolated from PCs and cardiac tube of *Mtpα*[KO] or *Mtpβ*[KO] mutants were comparable with that observed in flies reared on ND (Fig. 7 B). In contrast, loss of MTPα or MTPβ resulted in a remarkable drop in *upd3* expression when the flies were reared on HSD (Fig. 7 B). Given that *upd3* expression is restricted to the PCs (please refer to Fig. 2 A), these results connect increased FAO to upregulated *upd3* expression in the PCs of HSD-fed flies.

To ascertain whether the drop in *upd3* expression, as observed in the PCs of HSD-fed flies upon attenuating FAO, impacted *prc* expression in the fat cells, we analyzed the expression of *prc* in the fat cells of these flies. Knocking down the expression of *whd* in the PCs resulted in a drastic reduction in the levels of *prc* expression in the fat cells of HSD-fed flies, whereas *prc* expression remained unaltered in the flies reared on ND (Fig. 7 C and Fig. S4 B). A similar drop in fat-specific *prc* expression was detected when *whd* was knocked out in the PCs of HSD-fed flies (Fig. 7 C and Fig. S4 B). In agreement with these observations, a notable reduction in the amount of Prc accumulation around the heart of HSD-fed flies was observed upon downregulating *whd* in the PCs (Fig. 7, D–H). Finally, we analyzed the effect of

downregulating *whd* in the PCs on the cardiac function of HSD-fed flies. As evident from Fig. 7 I and Videos 6, 7, 8, and 9, the irregularity in heart beating pattern, as observed in HSD-fed flies, was significantly restored upon downregulating *whd* in the PCs. Accordingly, an appreciable rescue in the arrhythmia index was detected (Fig. 7 J). The heart period was partially restored when *whd* was either knocked down or knocked out (Fig. S4 C). However, restorations in diastolic interval (Fig. S4 D) and systolic interval (Fig. S4 E) were more apparent in cardiac tubes when *whd* was knocked out.

## JNK signaling acts upstream of FAO to regulate *upd3* expression in the PCs of HSD-fed flies

Given that either attenuating JNK signaling or downregulating FAO in the PCs of HSD-fed flies led to an appreciable suppression of the increased *upd3* expression, we wanted to determine the relation between these two pathways. To start with, the expression of TRE-dsRed in the PCs upon attenuating FAO was assayed. As evident from Fig. 8, A–E, the expression of TRE-dsRed remained unaltered in the PCs of HSD-fed flies upon downregulating *whd*. In a converse experiment, we analyzed the expressions of FAO-related genes when JNK signaling was downregulated by driving a dominant negative form of *bsk* in the PCs of HSD-fed flies. While a remarkable drop in *whd* transcripts was detected in the RNA isolated from PCs and cardiac tubes (Fig. 8 F), the expression of the rest of the genes coding for other FAO enzymes either exhibited a mild drop or no significant alteration upon loss of *bsk* from the PCs of HSD-fed flies (Fig. S4 F), indicating that the expression of *whd* is dependent on JNK signaling.

Next, we wanted to check the relation between JNK signaling and FAO in controlling *upd3* expression. Downregulating JNK signaling by expressing a dominant negative form of *bsk*, specifically in the PCs, resulted in a remarkable drop in *upd3* transcripts in the RNA isolated from PCs and cardiac tubes of flies reared on either normal or HSD (Fig. 8 G and Fig. S4 G). On the other hand, overexpression of *whd* in the PCs of flies reared on ND led to an increase in *upd3* transcripts as compared with control (Fig. 8 G and Fig. S4 G). Though not significant, a modest

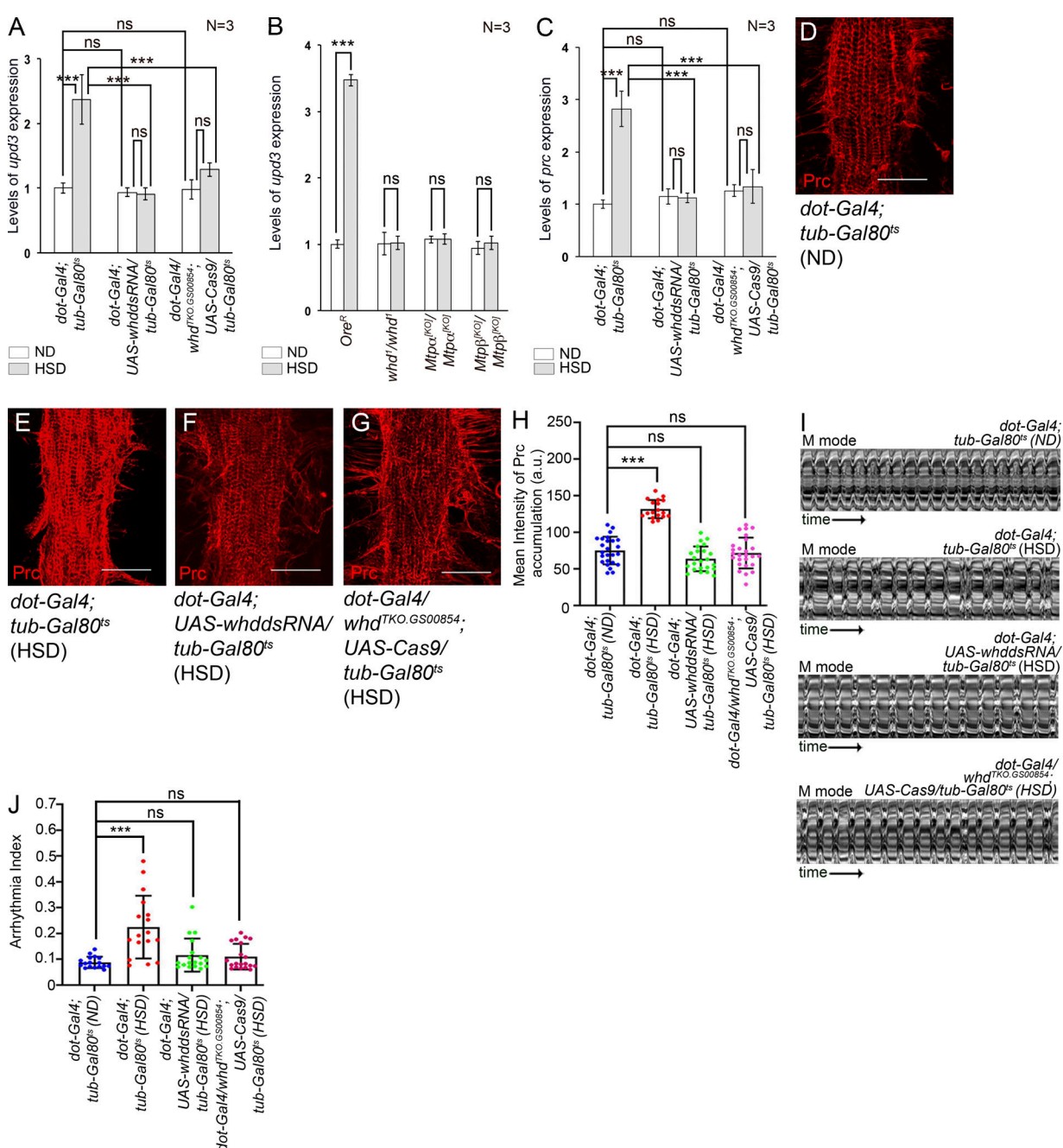

Figure 7. **Elevated levels of FAO contribute to the upregulation of *upd3* expression in the PCs of HSD-fed flies. (A)** Levels of *upd3* expression upon attenuating Whd activity in the PCs of ND or HSD-fed flies. The transcript levels are normalized to that of the constitutive ribosomal gene *rp49*. **(B)** Levels of *upd3* expression in the PCs of *whd*, *mtp-α*, and *mtp-β* homozygous mutant flies reared on either ND or HSD. The transcript levels are normalized to that of the constitutive ribosomal gene *rp49*. **(C)** Changes in the level of *prc* expression in the fat cells of ND or HSD-fed flies upon attenuating Whd activity in the PCs. The transcript levels are normalized to that of the constitutive ribosomal gene *rp49*. **(D–G)** Increased levels of Prc accumulation (red) around the heart of HSD-fed flies (E) as compared to that observed upon rearing on ND (D) get reduced upon attenuating Whd activity in the PCs (F and G). **(H)** Quantification of the mean fluorescence intensity for Prc accumulation around the second heart chamber of HSD-fed flies upon attenuating Whd function in the PCs. The dots represent the samples analyzed for each genotype. **(I)** Representative M-modes for heartbeats of adult flies showing the movement of the heart tube walls (y-axis) over time (x-axis). **(J)** Changes in arrhythmia index in the hearts of HSD-fed flies upon attenuating *whd* in the PCs. The dots represent the samples analyzed for each genotype. Genotypes are as mentioned. Data are represented as mean ± SD. P values (ns ≥ 0.05, ***P < 1 × 10⁻³) were obtained by two-way ANOVA with Tukey's multiple-comparison test (A, B, and C) or by one-way ANOVA with Tukey's multiple-comparison test (H and J). Scale bars, 50 µm.

increase in *upd3* transcripts was detected in the RNA isolated from PCs and cardiac tube of HSD-fed flies upon upregulating *whd* expression (Fig. 8 G). Subsequently, we checked for the levels of *upd3* transcripts by simultaneously upregulating *whd*

expression and downregulating JNK signaling in the PCs of HSD-fed flies. As shown in Fig. 8 G, the expected drop in the level of *upd3* transcripts because of attenuating JNK signaling got rescued due to overexpression of *whd*. Considering that the

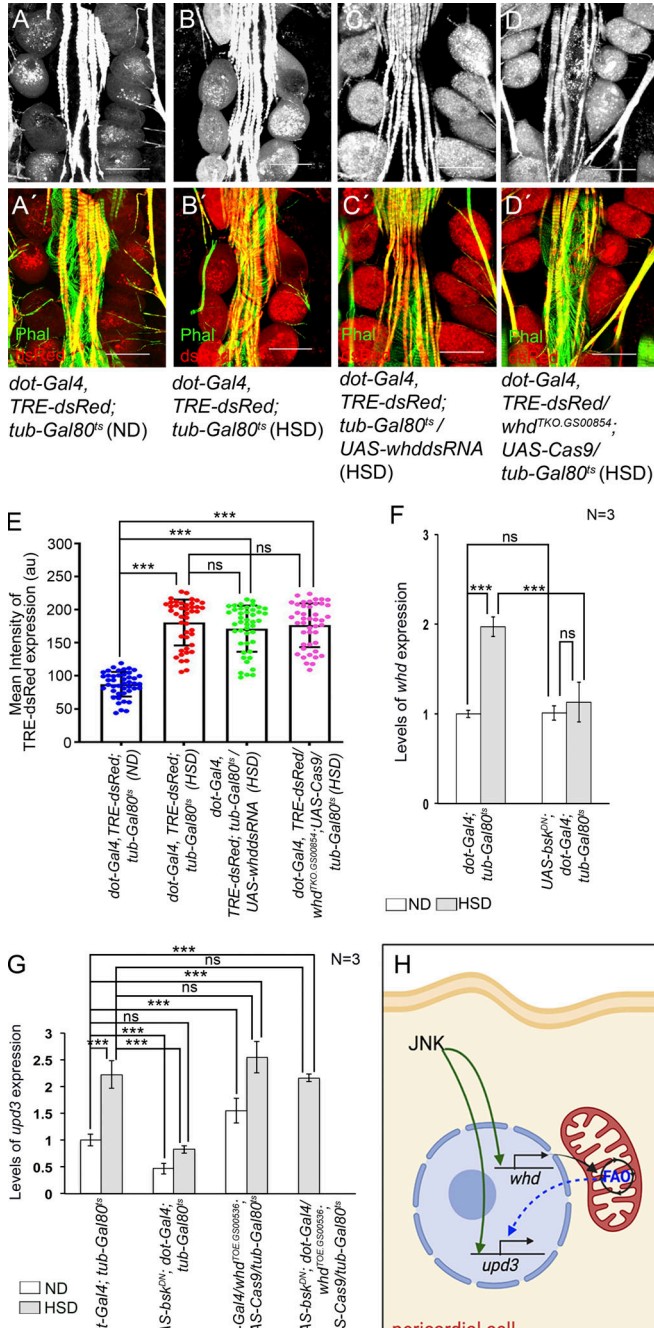

Figure 8. **JNK-dependent elevated levels of FAO upregulate *upd3* expression. (A–D')** Increase in the reporter TRE-DsRed expression (red) in the PCs of HSD-fed flies (B and B') as compared to that observed in flies reared on ND (A and A') remains unaltered even after attenuating Whd function in the PCs (C–D'). Phalloidin (green) marks the cardiac tube and the alary muscles. A, B, C, and D show TRE-dsRed expression in grayscale; A', B', C', and D' represent the merged images of the two colors. **(E)** Quantification of the mean fluorescence intensity for TRE-DsRed expression in the PCs of HSD-fed flies upon attenuating Whd function in the PCs. The dots represent the number of PCs analyzed for each genotype. **(F)** Changes in the level of whd transcripts in the RNA isolated from PCs and cardiac tube of ND- and HSD-fed flies upon attenuating JNK pathway in the PCs. The transcript levels are normalized to that of the constitutive ribosomal gene rp49. **(G)** Overexpression of whd remarkably recues the drop in pericardial upd3 expression of HSD-fed flies as observed upon attenuating JNK signaling in the PCs. The

expression of *upd3* is detected only in the PCs, but not in the cardiac tube, these results establish that JNK signaling, apart from its role as a transcriptional activator for *upd3*, acts upstream of FAO in regulating *upd3* expression in the PCs of HSD-fed flies (Fig. 8 H).

## FAO-dependent histone acetylation potentiates *upd3* expression in the PC of HSD-fed flies

Finally, we ventured to identify the mechanism underlying FAO-mediated upregulation of *upd3* expression in the PCs of HSD-fed flies. Besides serving as a substrate for the Krebs cycle, acetyl-CoA generated from FAO also plays an important role in the acetylation of various proteins, including histones, to modulate gene expression (Fig. 9 A) (McCarthy et al., 2018; Tiwari et al., 2020). Transfer of acetyl group from acetyl-CoA to conserved lysine residues on histones is achieved by histone acetyl transferases (HATs). Acetylation of lysine residues, in turn, neutralizes their positive charge, thereby causing histones to drift away from DNA. This released structure facilitates access to the transcriptional machinery leading to enhanced gene expression (Shvedunova and Akhtar, 2022). To ascertain whether increased histone acetylation contributes to upregulation of *upd3* expression in the PCs of HSD-fed flies, we first checked for the expression of the genes coding for HATs. Compared with the flies reared on ND, a remarkable enrichment in the transcript levels of the two HAT genes, *Gcn5* (Carré et al., 2005) and *chameau* (*chm*) (Grienenberger et al., 2002; Miotto et al., 2006), was observed in the RNA isolated from the PCs and cardiac tube of HSD-fed flies (Fig. 9 B). To document the status of histone acetylation in the PCs, we performed immunostainings with a battery of antibodies against acetylated H3K9, acetylated H3K18, and acetylated H3K27. As shown in Fig. 9, C, D″, and G, elevated levels of H3K9 acetylation were observed in the PCs of HSD-fed flies as compared with that observed in ND-fed flies. However, the levels of acetylated H3K18 (Fig. S5, A–C) and acetylated H3K27 (Fig. S5, D–F) remained unaltered in the PCs of HSD-fed flies when compared with that observed in the PCs of ND-fed flies. Increased levels of H3K9ac, but not H3K18ac or H3K27ac, suggest that high dietary sugar does not lead to increased acetylation of all lysine residues of histones. Instead, by site-specific histone acetylation, it generates a unique histone acetylation signature responsible for modulating gene expression in the PCs.

Next, we wanted to investigate whether the elevated levels of H3K9ac observed in the PCs of HSD-fed flies are an outcome of increased FAO. For that purpose, we downregulated the expression of *whd* in the PCs of HSD-fed flies and checked for the status of H3K9ac. PC-specific loss of *whd* resulted in significant suppression of the high levels of H3K9ac detected in the PCs of HSD-fed flies (Fig. 9, E–G). A similar drop in the levels of H3K9ac

transcript levels are normalized to that of the constitutive ribosomal gene rp49. **(H)** Signaling cascade triggered by JNK in regulating upd3 expression in the PCs of HSD-fed flies. Genotypes are as mentioned. Data are represented as mean ± SD. P values (ns ≥ 0.05, ***P < 1 × 10⁻³) were obtained by two-way ANOVA with correct with Tukey's multiple-comparison test (E, F, and G). Scale bars, 50 μm.

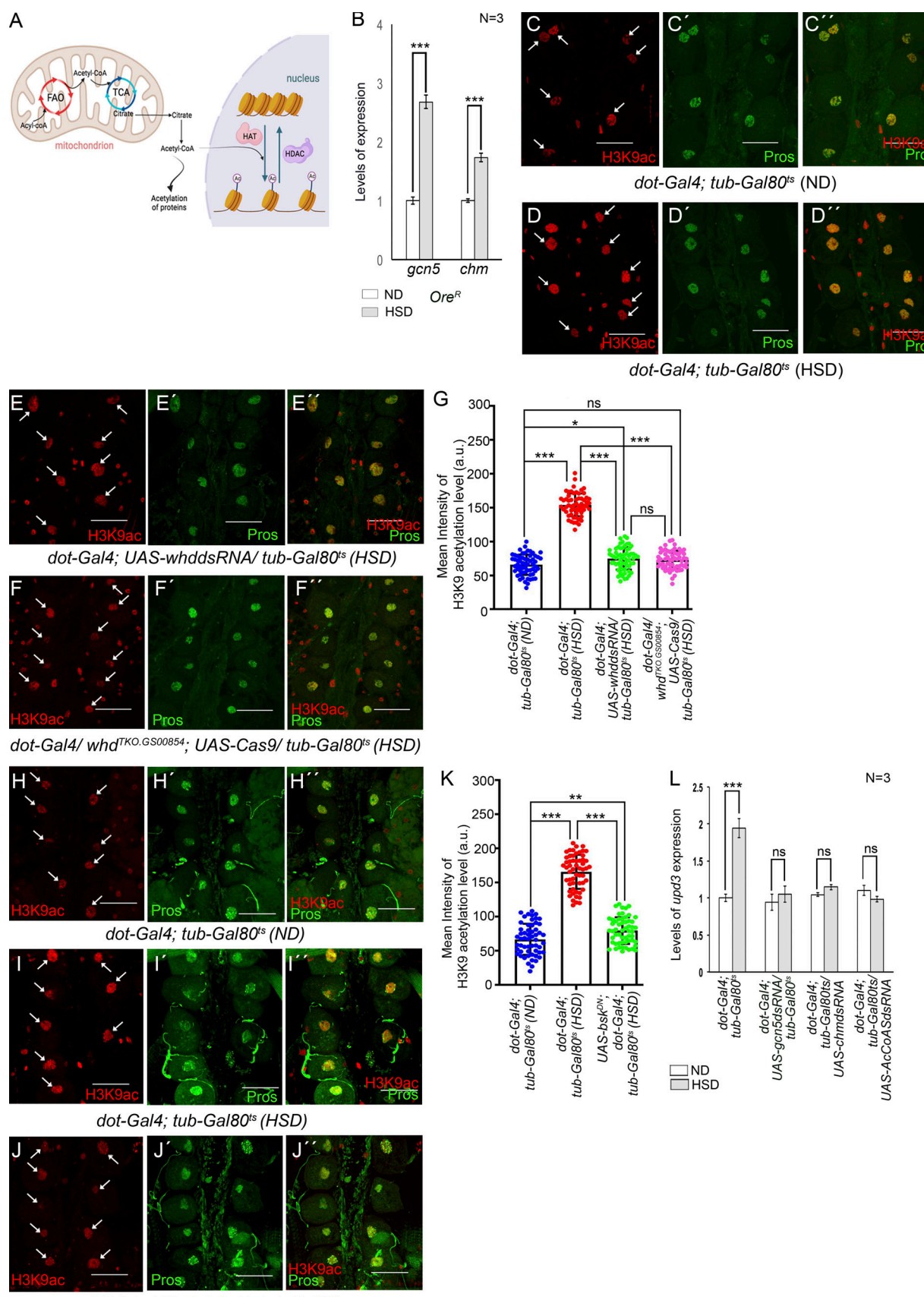

Figure 9. **FAO-induced histone acetylation leads to upregulated pericardial *upd3* expression. (A)** Schematic representation showing the connection between FAO and histone acetylation in a cell. HDAC, histone deacetylase. **(B)** Increase in the levels of *gcn5* and *chm*, the two genes that code for HAT, in the

PCs of HSD-fed flies. The transcript levels are normalized to that of the constitutive ribosomal gene *rp49*. **(C–D")** Compared to that observed in the PCs of ND-fed flies (C–C"), increased levels of H3K9ac (red) are observed in the PCs of HSD-fed flies (D–D"). Prospero (green) marks the PCs. C and D show the expressions of H3K9ac (red) in the PCs (marked with arrows), C' and D' show the expression of Prospero (green) in the same PCs, and C" and D" represent the merged images of the two colors. **(E–F")** Knocking down of *whd* (E–E") or knocking it out (F–F") in the PCs of HSD-fed flies suppresses the increase in the levels of H3K9ac (red) as observed in HSD-fed flies. Prospero (green) marks the PCs. E and F show the expressions of H3K9ac (red) in the PCs (marked with arrows), E' and F' show the expression of Prospero (green) in the same PCs, and E" and F" represent the merged images of the two colors. **(G)** Quantification of the mean fluorescence intensity for H3K9ac in the PCs of HSD-fed flies upon attenuating Whd function in the PCs. The dots represent the number of PCs analyzed for each genotype. **(H–J")** Compared with the PCs of ND-fed flies (H–H"), the increased levels of H3K9ac (red) as observed in the PCs of HSD-fed flies (I–I") experience an appreciable drop upon attenuation of JNK activity in the PCs of HSD-fed flies (J–J"). H, I, and J show the expressions of H3K9ac (red) in the PCs (marked with arrows), H', I', and J' show the expression of Prospero (green) in the same PCs, and H", I", and J" represent merged images of the two colors. **(K)** Quantification of the mean fluorescence intensity for H3K9ac in the PCs of HSD-fed flies upon attenuating JNK activity in the PCs. The dots represent the number of PCs analyzed for each genotype. **(L)** Independent downregulation of *gcn5*, *chm*, and *AcCoAS* in the PCs of HSD-fed flies leads to a remarkable drop in *upd3* expression. The transcript levels are normalized to that of the constitutive ribosomal gene *rp49*. Genotypes are as mentioned. Data are represented as mean ± SD. P values (ns ≥ 0.05, *P < 0.05, **P < 0.01, ***P < 1 × 10⁻³) were obtained by unpaired Student's *t* test (two-tailed) with Welch's correction (B) or by two-way ANOVA with Tukey's multiple-comparison test (G, K, and L). Scale bars, 50 μm.

was detected upon attenuating JNK signaling by expressing a dominant negative form of *bsk* in the PCs of HSD-fed flies (Fig. 9, H–K). Together, these results connect high dietary sugar–induced JNK-FAO axis with high levels of H3K9 acetylation in the PCs.

Finally, we wanted to determine whether the elevated levels of HATs expressed in the PCs of HSD-fed flies contribute to upregulating *upd3* expression. Independent downregulation of the HAT genes, *Gcn5* and *chm*, in the PCs of HSD-fed flies resulted in a remarkable restoration in the elevated levels of *upd3* expression (Fig. 9 L and Fig. S5 G) associated with the PCs of flies reared on high dietary sugar. In an alternate approach to decipher the link between histone acetylation and elevated levels of *upd3* expression, we specifically knocked down the expression of *acetyl-CoA synthase/AcCoAS* (the *Drosophila* ortholog of ACSS2) (Mews et al., 2017) in the PCs of HSD-fed flies and checked for *upd3* expression. AcCoAS is a crucial enzyme that catalyzes the conversion of free acetate into acetyl-CoA and in doing so, influences gene expression by regulating the acetylation of various proteins including histones (Ling et al., 2022; Mews et al., 2017). Downregulation of *AcCoAS* in the PCs of HSD-fed flies resulted in a remarkable drop in the elevated levels of *upd3* expression (Fig. 9 L and Fig. S5 G) comparable with that observed upon downregulation of *whd* and HAT genes. Put together, these molecular and genetic analyses confirm the essential role of FAO-dependent acetylation of histones in *upd3* upregulation in the PCs of HSD-fed flies.

## Discussion

A schematic of the inter-organ signaling cascade linking increased FAO in the PCs of HSD-fed flies to upregulation of *prc* expression in the fat cells leading to cardiac dysfunction resulting from excess Prc accumulation in the cardiac ECM, unraveled primarily based on in vivo loss-of-function genetic analysis, is shown in Fig. 10. We conclude that consumption of high dietary sugar leads to altered lipid metabolism and JNK-dependent upregulation of FAO in the PCs. Elevated levels of FAO, in turn, potentiate *upd3* overexpression by modulating the acetylation state of H3K9. Eventually, overexpression of pericardial *upd3* induces excessive Prc synthesis from the fat cells, leading to progressive cardiac fibrosis. Genetic manipulations of

members of this newly identified pathway in flies reared on high dietary sugar prevent upregulation of *upd3* in the PCs and help partially rescue the cardiac defects associated with excessive Prc accumulation in the cardiac ECM. Though the reason behind increased JNK activation in the PCs of HSD-fed flies is unclear, the results establish that the ROS-Ask1-JNK/p38 signaling cascade, implicated in *upd3* regulation during normal physiological conditions, does not play any role in HSD conditions. Instead, the outcome of this study unravels a two-tier regulatory mechanism by which JNK controls *upd3* expression in the PCs of HSD-fed flies. Besides its instructive role as a transcriptional activator of *upd3*, JNK elicits a permissive role via FAO-dependent modification of the epigenetic landscape to facilitate *upd3* expression. Intriguingly, the PCs of HSD-fed flies exhibit a unique histone acetylation signature with increased H3K9ac and unaltered H3K18ac/H3K27ac compared with the flies reared on ND. Of note, several studies in the recent past have documented the dietary influence on site-specific histone acetylation in different organs. For instance, high-fat feeding in mice has been

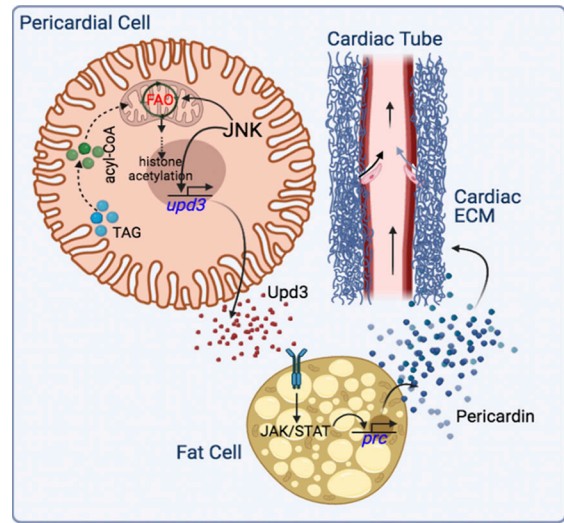

Figure 10. **Model showing the inter-organ communication circuitry that connects JNK, FAO, and Upd3 in the PCs to upregulate Prc expression in the fat cells of HSD-fed flies leading to cardiac dysfunction due to increased accumulation of Prc in the cardiac ECM.** Arrows in the cardiac tube mark the directionality of hemolymph flow.

associated with reduced H3K18ac and H3K23ac levels in white adipose tissue and the pancreas (Carrer et al., 2017) while hepatic gene transcription regulation is associated with changes in H3K27ac levels (Siersbæk et al., 2017). Furthermore, in type 2 diabetic mice, progressive glomerulosclerosis is associated with renal histone H3K9 and H3K23 acetylation (Sayyed et al., 2010).

Recent studies involving both fly and mouse models have attributed flux through the hexosamine biosynthetic pathway to diet-induced cardiac dysfunction. In flies, increasing hexosamine flux phenocopies HSD-mediated heart dysfunction and genetic attenuation of the hexosamine biosynthetic pathway, specifically in the adult heart, leads to improved heart function of HSD-fed flies. Though the underlying mechanism is not evident, at least it is clear that the restoration in cardiac function is not associated with any detectable alteration in the elevated levels of Prc accumulation observed in HSD-fed flies (Na et al., 2013). In contrast, it has been demonstrated that elevated glucose concentration induces arrhythmias in murine cardiac myocytes by autonomous O-linked glycosylation-dependent activation of CaMKII (Erickson et al., 2013). Intriguingly, a high dietary sugar–induced hexosamine pathway has also been implicated in the etiology of PC dysfunction in adult flies for its role in epigenetic modification–induced altered gene expression. Activation of the Polycomb repressor complex due to O-GlycNAcylation of the PRC1 subunit Polyhomeotic (Ph) impacts nuclear gene expression and thereby links increased dietary sugar to loss of the Nephrin ortholog Sns in the PCs, leading to reduced nephrocytic function (Na et al., 2015). By demonstrating the role of JNK-FAO-H3K9Ac axis in upregulating upd3 expression in the PCs of HSD-fed flies, the outcome of our study not only provides evidence for an alternate epigenetic mechanism by which high dietary sugar modulates gene expression in the PCs but also braids with the previous observations in providing a better understanding of high dietary sugar–induced diabetic nephropathy and cardiac dysfunction. An increased rate of FAO performs two distinct roles in regulating gene expression in the PCs of HSD-fed flies. On the one hand, it promotes hexosamine pathway-dependent O-GlycNAcylation of Ph by supplying acetyl-CoA, essential for the hexosamine biosynthetic pathway; on the other hand, acetyl-CoA is utilized for histone acetylation to modulate upd3 expression. While the first process exacerbates the nephrocytic function of the PCs, the second one aggravates cardiac dysfunction by augmenting excessive accumulation of Prc around the cardiac tube. Furthermore, our results, in conjunction with the previous findings, elucidate that activation of the JNK-FAO-H3K9Ac axis in the PCs of HSD-fed flies enhances cardiac dysfunction by increased Prc accumulation around the heart, whereas escalated hexosamine flux acts on other specific, yet unknown, components of the heart to induce cardiac dysfunction of HSD-fed flies. This might be why the increase in Prc accumulation around the cardiac tube of HSD-fed flies remained unaltered even after heart-specific attenuation of the hexosamine pathway.

Cytokines are pleiotropic molecules that modulate a broad spectrum of biological processes in a highly context-dependent and cell-type-specific manner. Although cytokines are traditionally studied concerning immune responses, many findings have recognized their role in modulating lipid metabolism in cancer and metabolic disorders. Numerous inflammatory cytokines including IL-17A, IL-6, IL-4, TNF-a, IL-15, IL-1, IL-32, and IL-33 have been implicated in modulating the levels of lipolysis (Fuster et al., 2011), lipogenesis (Tsao et al., 2014; Zúñiga et al., 2010), FAO (Carey et al., 2006; Stienstra et al., 2010), cholesterol accumulation (Isoda et al., 2005; Xu et al., 2017), and triglyceride storage (Kralisch et al., 2005) to either promote or inhibit the development of several metabolic diseases. Likewise, specific cytokines such as IL-6, TNF-a, IL-15, and IL-1 have been postulated to promote or inhibit cancer cachexia and prostate cancer by affecting several aspects of lipid metabolism (Batista et al., 2012; Han et al., 2018; Rohena-Rivera et al., 2017). However, to our knowledge, any evidence supporting FAO-mediated activation of cytokine expression is yet to be reported. By providing in vivo genetic evidence for activation of the cytokine Upd3 by FAO, our results shed light on a new facet that integrates cytokine biology and lipid metabolism. Given that the underlying mechanism involves the highly conserved process of histone acetylation, this regulation can modulate cytokine expression in different physiological contexts.

Excessive lipid accumulation and altered fatty acid metabolism are increasingly recognized as critical pathogenic processes contributing to the severity of diabetic cardiomyopathy. The increased reliance of the diabetic heart on fatty acids negatively impacts cardiac efficiency (Lopaschuk et al., 2021; Mazumder et al., 2004; Randle et al., 1963), impairs insulin metabolic signaling (Atkinson et al., 2003; Samuel and Shulman, 2016), elevates the production of ROS (Tsushima et al., 2018), increases mitochondrial membrane uncoupling, promotes lipotoxicity (Finck et al., 2002; Zhou et al., 2000), and exacerbates diabetic cardiomyopathy. As a result, the attenuation of cardiac FAO represents a potential therapeutic target for treating diabetic cardiomyopathy (Fragasso et al., 2006; Saeedi et al., 2005; Yue et al., 2005). However, even though the development of myocardial fibrosis, characterized by the uncontrolled accumulation of collagen and fibronectin, promotes cardiac stiffness and is a cardinal component of diabetic cardiomyopathy, the contribution of elevated FAO in diabetic cardiac fibrosis, if any, has not been evidenced. On a similar note, although epigenetic histone modifications profoundly influence fibroblast gene transcription in injured tissues (Schuetze et al., 2014; Zhang et al., 2017), the mechanistic underpinnings that link metabolic perturbations and epigenetic modifications to trigger fibrogenic activation are obscure. From these perspectives, our findings provide in vivo evidence for a mechanism that connects high dietary sugar-induced accelerated FAO to histone acetylation–dependent upregulation in cytokine expression that potentiates progressive cardiac fibrosis by increased expression of the collagen-like molecule Prc. Investigating whether the pathomechanism identified here also applies to fibrotic disorders in higher vertebrates would be imperative.

## Materials and methods

### *Drosophila* stocks and fly husbandry
All the fly stocks were routinely maintained at 25°C (if not otherwise mentioned) on a standard cornmeal-yeast-agar food

containing 0.15 M sucrose. For experimental purposes, freshly eclosed flies were reared on ND (standard cornmeal-yeast-agar food with 0.15 M sucrose) for 2 days. Thereafter, while the control flies were reared on ND for the rest of their life, the experimental batches of flies were reared on HSD (standard cornmeal-yeast-agar food with a final concentration of 1.0 M sucrose). 100 ml food contained maize powder (5.1 g), agar (0.9 g), yeast (1.8 g), methyl paraben (0.3 g), propionic acid (300 μl), and sucrose (5.13 g = 0.15 M for ND and 34.2 g = 1 M for HSD). 20-day-old adult female flies were used for all the analyses. To induce transgene expression using Gal4 drivers, flies of different genetic backgrounds were maintained at 18°C till eclosion. 2-day-old freshly eclosed flies were then reared at 29°C for 14 days (equivalent to 18 days at 25°C) for analyses. The different fly strains used in this study are provided in Table S1.

### Immunostaining of adult cardiac tissue and PCs

All immunostaining experiments were carried out at least three times in accordance with the standard protocol. Adult female abdomens were dissected on ice in 1× phosphate-buffered saline (PBS) and fixed overnight at 4°C in 10% paraformaldehyde. The samples were washed the next day with PBS and all the adnominal fat bodies were removed so that the cardiac tissue and the PCs could be properly exposed to all the chemical treatments. The samples were then fixed in 5% paraformaldehyde for about 2 h at room temperature before being washed with 0.3% PBT (0.3% Triton X-100 in 1X PBS) for 90 min. Tissues were then incubated in primary antibody for 24 h diluted in PAXDG (0.3% PBT with 0.3% sodium deoxycholate, 5% normal goat serum, and 1% bovine serum albumin) at 4°C. Following washes in 0.3% PBT, the samples were incubated for 24 h in secondary antibody in PAXDG at 4°C. The tissues were washed in PBT and stained with phalloidin (1:500 in PBS) at 4°C for 48 h followed by staining with 4′,6-diamidino-2-phenylindole at room temperature for 30 min.

For immunostaining with antibodies against acetylated histones, a slight modification was done in the above-mentioned protocol. The adult tissues were dissected in 1X PBS supplemented with deacetylase inhibitors: 10 mM sodium butyrate (EMD 19-137; Millipore) and 10 mM nicotinamide (#72345; Sigma-Aldrich). The rest of the protocol remained the same as already mentioned.

The samples were mounted in vectashield and immunofluorescence images were captured in a Leica SP8 confocal microscope under identical conditions (laser power, gain, and scan settings). The images were processed in parallel and analyzed under the same conditions using Fiji software. The following primary antibodies were used: mouse anti-Prc (1:10; Developmental Studies Hybridoma Bank EC11), mouse anti-Prospero (1:20; Developmental Studies Hybridoma Bank MR1A), rabbit anti-phospho p38 (1:100; Thr[180]/Tyr[182], clone 3D7; Cell Signaling Technology), mouse anti-β galactosidase (1:100; no. Z3781; Promega), rabbit anti-H3K9ac (Cat#9927; Cell Signaling Technology), rabbit anti-H3K18ac (Cat#9927; Cell Signaling Technology), and rabbit anti-H3K27ac (Cat#9927; Cell Signaling Technology).

### Detection of ROS

ROS detection with DHE dye (Molecular Probes no. D11347; Invitrogen) was carried out as previously described in (Owusu-Ansah and Banerjee, 2009). Adult female abdomens were dissected in 1X PBS at room temperature followed by incubation in 30 μM DHE solution in 1X PBS for 15 min at room temperature under dark conditions. After DHE incubation, tissues were washed with 1X PBS for 5 min before being fixed with 10% paraformaldehyde for 10 min and again washed with 1X PBS for 5 min at room temperature in dark. Finally, the samples were immediately mounted and imaged using a Leica SP8 confocal microscope (Leica Application Suite X acquisition software) under identical conditions (laser power, gain, and frame size). The images were analyzed and processed using Fiji software under the same conditions.

### Bodipy/LipidTOX staining

Adult female abdomen were dissected on ice in 1X PBS and fixed with 10% paraformaldehyde in 1X PBS overnight at 4°C. The following day, the tissues were washed with 1X PBS and the abdominal fat body cells were removed, especially around the PCs, for proper penetration of the stain. Samples were again fixed with 10% paraformaldehyde for 1 h followed by washes in 0.3% PBT (0.3% Triton-X in 1X PBS) for 2 h (8 washes of 15 min each) at room temperature. Then, the tissues were washed twice with 1X PBS before incubating in Bodipy (1:500 in 1X PBS) or LipidTOX solution (1:500 in 1X PBS) for 30 min, followed by washing with 1X PBS. The processed samples were immediately mounted in vectashield, imaged using Laser Scanning Confocal Microscope (Zeiss LSM 900; ZEN 3.1 acquisition software), and processed using Fiji software.

### Nile Blue A staining

Adult female abdomens were dissected on ice in 1X PBS and fixed with 10% paraformaldehyde in 1X PBS overnight at 4°C. The following day, the tissues were washed with 1X PBS and the abdominal fat body cells were removed, especially around the PCs, for proper penetration of the stain. Samples were again fixed with 10% paraformaldehyde for 1 h followed by washes in 1X PBS for 45 min (three washes of 15 min each) at room temperature. Then, the tissues were incubated in Nile Blue (5 μM in 1X PBS) (Cat. No. #N0766; Sigma-Aldrich) solution for 15 min, followed by three washes with 1X PBS. The processed samples were immediately mounted in vectashield, imaged using Laser Scanning Confocal Microscope (Zeiss LSM 900; ZEN 3.1 acquisition software), and processed using Fiji software.

### 2-NBDG staining

Adult female abdomens were dissected in Schneider's medium at room temperature and immediately incubated in 0.25 mM 2-NBDG solution (Cat. No. # N13195; Invitrogen) for 45 min at room temperature under dark conditions as prescribed in de la Cova et al. (2014). After incubation, followed by a quick PBS wash, the tissues were fixed in precooled 10% paraformaldehyde prepared in 1X PBS on ice for 1.5 h. Then the tissues were washed with chilled 1X PBS for 5 min, immediately mounted in vectashield, imaged using Laser Scanning Confocal Microscope (Zeiss LSM 780; ZEN, 2010 B SP1 acquisition software), and processed using Fiji software.

## RNA isolation from adult cardiac tissue and fat body

To study the expression levels of genes in the PCs, the PCs along with the adult cardiac tubes were carefully dissected out from around 40 to 50 female flies in 1X PBS. Care was taken to have the cardiac tube and the associated PCs only. 1X PBS was replaced with TRIzol (Ambion no. 15596018) and RNA was isolated according to the manufacturer's instructions.

To study the expression of *prc* in the fat cells, abdominal cuticles (with fats, cuticular muscles, and cuticle) from 25 to 30 adult flies were dissected in Schneider's insect medium on ice. Only the fat bodies were carefully dissected from these abdominal tissues and collected in a 1.5-ml microcentrifuge tube with Schneider's insect medium on ice. The samples were then centrifuged for 5 min at 5,000 RPM (4°C) to settle down the fat cells at the bottom of the tube. Then the samples were homogenized in Schneider's insect medium and the lysate was passed through a 22G syringe (BD No. 309631) two to three times. The collected lysate was centrifuged for 5 min at 5,000 RPM (4°C) to pellet down the cells. The Schneider's insect medium was replaced with TRIzol (Ambion no. 15596018) and RNA was isolated according to the manufacturer's instructions.

cDNA was generated using the Verso cDNA synthesis kit (Molecular Probes no. AB1453A) according to the manufacturer's instructions. The Bio-Rad CFX96 Touch Real-Time PCR Detection System was used to perform real-time quantitative polymerase chain reaction (RT-qPCR) using the TB Green Premix Ex Taq (Tli RNaseH Plus) (no. RR440A; Takara). Graded RT-qPCR was used to determine the optimal primer melting temperature for each primer set. The 2-δCt method was used to determine the relative levels of mRNAs. The transcript levels were normalized to those of the constitutive ribosomal gene coding for ribosomal protein 49 (rp49). Three to four independent cDNA samples were used in each RT-qPCR experiment. The figures depict the total data from all samples. Primers were created in such a way that they span the exon-exon junction. Table S2 provides the list of primers used in our RT-qPCR experiments.

## Insulin sensitivity assay from whole abdomen of *Drosophila*

For insulin sensitivity assay, adult female flies were dissected open in Schneider's medium supplemented with 10% FBS to completely expose the body cavity to the following chemical treatments. 5 μM recombinant human insulin (Cat # I1376497001; Roche) was added to the Schneider's medium and samples were incubated at room temperature for 20 min. For the control samples, dilution buffer (10 mM HEPES) was added to the medium. The rest of the treatments were the same for both the sample groups.

Once the incubation was done, the heads were removed and the bodies were homogenized on ice using an autoclave pestle in 1% Triton X-100 in 1X PBS buffer (pH 7.2) supplemented with 1% mammalian Protease Inhibitor Cocktail (P8340; Sigma-Aldrich) along with phosphatase inhibitor. The homogenate was centrifuged at 20,000 $g$ for 15 min at 4°C and the supernatant was mixed with an equal volume of 2× Laemmli Buffer (100 mM tris-HCl [pH 6.8]; 4% SDS: Sigma-Aldrich, catalog no. L3771; 0.2% Bromophenol Blue, HIMEDIA, catalog no. MB123; 20% glycerol and 200 mM β-mercaptoethanol, catalog no. M3148; Sigma-Aldrich). Finally, the protein solution was boiled at 95°C for 5 min

before it was ready to use for further analysis. The Bradford Assay (catalog no. 5000006; Bio-Rad) was used to quantify the protein concentration. 1 μl of protein sample was mixed with 1 ml of five-times-diluted Bradford reagent, incubated at room temperature for 5 min, and the OD of the solution was measured at 595 nm in a spectrophotometer. The concentration of protein was calculated using a freshly prepared standard curve. The primary antibodies used for western blot analysis are as follows: mouse anti-Tubulin (1:1,000; T6199; Sigma-Aldrich) and rabbit anti-active pAkt (1:1,000; Cat # 4054; Cell signaling). For secondary antibodies, horseradish peroxidase (HRP)–conjugated anti-mouse (no. A00160; GenScript) and HRP-conjugated anti-rabbit (no. A00098; GenScript) were used at 1:5,000 dilution. The western blots were developed using Luminata Crescendo Western HRP substrate (Millipore). The band intensities were analyzed using Fiji software and normalized with tubulin bands that served as loading controls. Western blot analyses were conducted on three to four independent protein samples per condition. The graphs represent the combined data from all samples.

## Fly heartbeat analysis

For all the cardiac functional analyses, semi-intact female *Drosophila* were used and the cardiac contractility was measured as previously described in Fink et al. (2009). High-speed digital movies of beating hearts were captured under bright-field using QImaging optiMOS scMOS camera R696-M-16-C with a 20× water immersion lens (Carl Zeiss). The neighboring PCs and fat tissue were removed during sample preparation to clearly access the cardiac tube for video recording. The movies were captured at 100 fps for a duration of 10 s. M-modes and quantitative data were generated using SOHA, a MATLAB-based image analysis software (Fink et al., 2009). M-mode of a particular sample is the snapshot of the movement of heart edges over time. Heart periods were defined as the time between the ends of two consecutive diastolic intervals, and arrhythmia index was quantified as the standard deviation (SD) of all heart periods in each record normalized to the median heart period for each fly.

## Life-span analysis

The life-span analysis experiments were carried out with three to four replicates of 60 flies (40 females and 20 males) per genotype. Every day, the flies were transferred to new food bottles. When the flies were flipped, dead female flies in each bottle were counted and removed. Female survival curves for each genotype ($n$ = 120–160 flies, three to four independent replicates) were generated in GraphPad Prism version 6.00 using a log-rank Mantel–Cox test.

## Intensity analysis of images

For intensity analyses of DHE, phospho-p38, HexA-GFP, pgi-GFP, Glut1-GFP, scully-YFP, CG3902-YFP, 2-NBDG, LipidTOX, Bodipy, Hnf4-Gal4UASmcd8GFP, TRE-dsRed, and Nile Blue A, the PCs of the control and experimental samples were marked, and the mean fluorescence intensity (mean gray pixel intensity of the desired area of maximum intensity projections of the Z stacks) was measured using Fiji software. In case of immunostainings with antibodies against acetylated histones, the nuclei of the PCs were marked and mean fluorescence

intensity was measured using Fiji software. A total of 50–70 PCs from 12 to 15 individual flies were analyzed for each genotype/dietary alteration. In particular, the PCs around the second chamber of the cardiac tube were analyzed for the sake of uniformity. To determine the intensity of Prc accumulation around the second chamber of the cardiac tube, the entire second chamber was marked and the mean fluorescence intensity (mean gray pixel intensity of the desired area of maximum intensity projections of the Z stacks) was measured using Fiji software. Data were obtained from 20 individual flies for analyzing the intensity of Prc accumulation around the second chamber of the cardiac tube. For all the genotypes, the samples were processed in parallel and all the analyses were done under identical conditions. From these readings, the mean gray values (for the same size of region of interest) of the background were deducted to obtain the mean gray value minus blank. The final intensity values were plotted using GraphPad Prism version 6.00, where the dots represent the number of PCs/cardiac chambers analyzed and the error bars represent SD.

### Quantification and statistical analysis

For all the statistical analyses, GraphPad Prism 6 software was used. The analyses of the data were performed by using either unpaired Student's $t$ test (two-tailed) with Welch's correction or one-way/two-way analysis of variance (ANOVA) followed by Tukey's multiple comparison test. For the life-span data, Kaplan–Meier survival curves were generated with GraphPad Prism 6 software and curve comparison was performed using Log-Rank (Mantel–Cox) test. All of the survival graphs shown are the result of three to four independent repeats of 40 flies per genotype. Statistical significance was accepted with P values of <0.05; <0.01; and <0.001, mentioned as *, **, and ***, respectively.

### Online supplemental material

Fig. S1 shows the hallmark features of T2D exhibited by the flies reared on HSD including an increase in *prc* expression. Fig S2 shows the changes in the levels of *upd3* and *prc* expressions in the control lines when reared on HSD, changes in heart period, diastolic and systolic intervals upon knocking down *upd3* in the PCs of HSD-fed flies, and drop in elevated levels of physiological ROS in the PCs of HSD-fed flies. Fig. S3 shows altered lipid metabolism in the PCs of HSD-fed flies and changes in the levels of *upd3* expressions in the control lines when reared on HSD. Fig. S4 shows the changes in the levels of *upd3* and *prc* expressions in the control lines when reared on HSD, changes in heart period, diastolic and systolic intervals upon either knocking down or knocking out of *whd* in the PCs of HSD-fed flies, and expression levels of genes involved in FAO upon attenuating JNK signaling in the PCs of HSD fed flies. Fig. S5 shows the levels of H3K18ac and H3K27ac in the PCs of HSD-fed flies and the expression levels of upd3 in the control lines when reared on HSD. Table S1 shows the details (genotypes and sources) of the *Drosophila* stocks used in this study. Table S2 has the list of primers and their sequences used for this study. Videos 1 and 2 show the beating second chamber of the cardiac tube (heart) of adult *Drosophila* (*Ore^R*) reared on ND and HSD, respectively. These movies are related to Fig. 1 H. Videos 3 and 4 show the beating second chamber of the heart of adult flies (*dot-Gal4*; *tub-Gal80^ts*) reared on ND and HSD, respectively. Video 5 shows the beating second chamber of the heart of adult flies (*dot-Gal4*; *tub-Gal80^ts*/*UAS-upd3dsRNA*) reared on HSD. Videos 3, 4, and 5 are related to Fig. 2 J. Videos 6 and 7 show the beating second chamber of the heart of adult flies (*dot-Gal4*; *tub-Gal80^ts*) reared on ND and HSD, respectively. While Video 8 shows the beating second chamber of the heart of adult flies (*dot-Gal4*; *UAS-whddsRNA/tub-Gal80^ts*) reared on HSD, Video 9 represents the beating second chamber of the heart of adult flies (*dot-Gal4/whd^TKO.GS00854*; *UAS-Cas9/tub-Gal80^ts*) reared on HSD. Videos 6, 7, 8, and 9 are related to Fig. 7 I.

### Data availability

All data underlying the research present in this manuscript are available from the corresponding author upon request (sudip@iisermohali.ac.in).

## Acknowledgments

We thank E. Bach (New York University Grossman School of Medicine, New York, NY, USA), J.A. Hoffmann (University of Strasbourg, Strasbourg, France), S. Newfeld (Arizona State University, Tempe, AZ, USA), U. Banerjee (University of California, Los Angeles, Los Angeles, CA, USA), Irene Miguel-Aliaga (The Francis Crick Institute, London, UK), and D. Bohmann (University of Rochester Medical Center, Rochester, NY, USA) for reagents; Bloomington *Drosophila* Stock Center and Kyoto Stock Center *Drosophila* Genetic Resource Center for fly stocks; Developmental Studies Hybridoma Bank for antibodies; other members of the laboratory for critical input; and Indian Institute of Science Education and Research Mohali (IISER Mohali) confocal (Leica SP8; Zeiss LSM900 and Zeiss LSM780 systems) microscopy facility for imaging. The models were created using BIORENDER.

This study was funded by IISER Mohali (J. Gera, G. Chauhan, A. Choudhary, L. Rani), Indian Council of Medical Research, India (3/1/3/JRF-2020/HRD(LS)/132109/46) (D. Kumar), Science and Engineering Research Board, Department of Science and Technology, India (CRG/2020/000511) (S. Mandal), and Wellcome Trust DBT India Alliance, India (IA/S/17/1/503100) (L. Mandal). Open Access funding provided by IISER Mohali.

Author contributions: J. Gera: Formal analysis, Investigation, Visualization, Writing—review and editing, D. Kumar: Investigation, Visualization, G. Chauhan: Investigation, Visualization, A. Choudhary: Investigation, Visualization, L. Rani: Investigation, L. Mandal: Funding acquisition, Visualization, Writing—original draft, Writing—review and editing, S. Mandal: Conceptualization, Formal analysis, Funding acquisition, Methodology, Project administration, Supervision, Visualization, Writing—original draft, Writing—review and editing.

Disclosures: The authors declare no competing interests exist.

Submitted: 21 June 2023

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

# Supplemental material

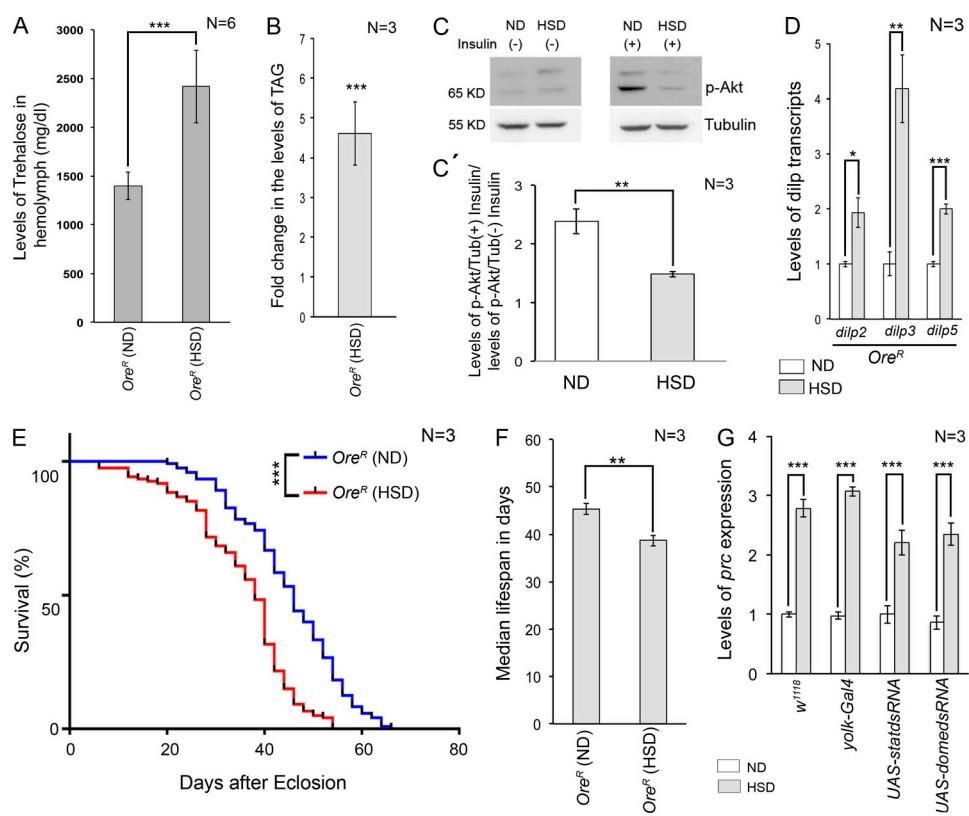

Figure S1.  **Flies fed on high dietary sugar exhibit hallmark features of T2D and excessive *prc* expression. (A)** Increase in hemolymph trehalose levels in flies reared on HSD compared with those reared on ND on day 20AE. **(B)** Increased levels of TAG in flies reared on HSD on day 20AE. **(C and C')** Changes in the levels of p-Akt in flies reared on ND and HSD with or without insulin treatment. **(D)** Increase in the levels of *dilp2*, *dilp3*, and *dilp5* expressions in flies reared on HSD compared with those reared on ND on day 20AE. **(E)** Survival rates of flies reared on either ND or HSD. Flies fed on HSD exhibited reduced lifespan. **(F)** The median lifespan of HSD-fed flies was 7 days shorter than that of the flies reared on ND. **(G)** Feeding on HSD led to an increase in the levels of *prc* expression in the PCs of the flies of all the genotypes mentioned. The transcript levels are normalized to that of the constitutive ribosomal gene *rp49*. Genotypes are as mentioned. Data are represented as mean ± SD. P values (*P < 0.05, **P < 0.01, ***P < 1 × 10$^{-3}$) were obtained by unpaired Student's *t* test (two-tailed) with Welch's correction (A, B, C', D, and F), by two-way ANOVA with Tukey's multiple-comparison test (G), and curve comparison was performed using Log-Rank (Mantel-Cox) test (E). Source data are available for this figure: SourceData FS1.

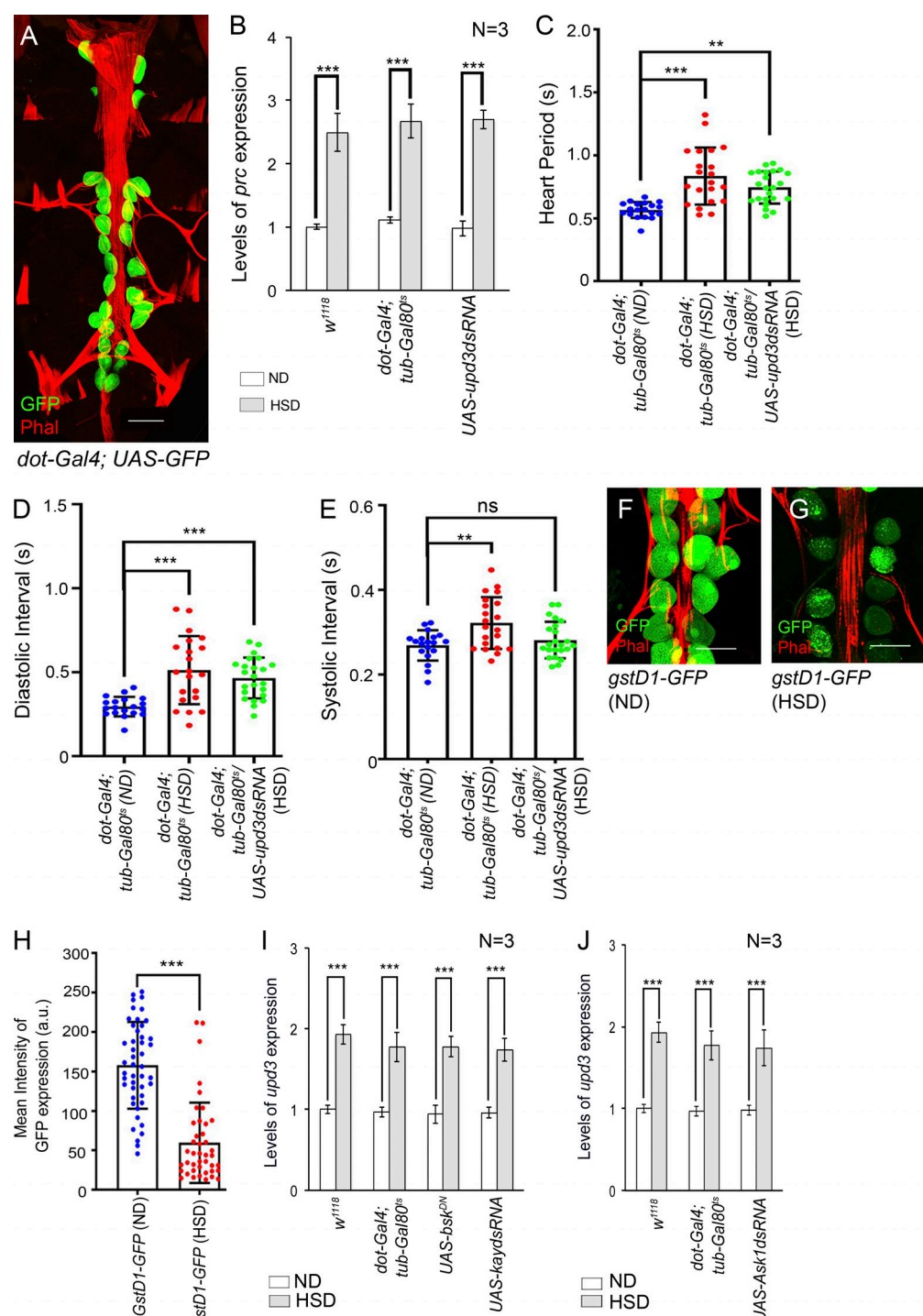

Figure S2. **PC-specific downregulation of *upd3* expression and JNK signaling result in a rescue in the elevated levels of *prc* expression in the fat cells of HSD-fed flies. (A)** Expression of *dot-Gal4* driving *UAS-GFP* (green) specifically in the PCs adjacent to the cardiac tube (red; stained with phalloidin). The abdominal body wall muscles and the alary muscles are also stained with phalloidin red. **(B)** Increase in the levels of *prc* expression in fat cells of the flies of the genotypes mentioned when reared on HSD. The transcript levels are normalized to that of the constitutive ribosomal gene *rp49*. **(C–E)** Changes in heart period (C), diastolic interval (D), and systolic interval (E) upon knocking down *upd3* in the PCs of HSD-fed flies. The dots represent the samples analyzed for each genotype. **(F and G)** Drop in the reporter GFP expression for *gstD1* in the PCs of HSD-fed flies (G) as compared to those reared on ND (F). Phalloidin (red) marks the cardiac tube and the alary muscles. **(H)** Quantification of the mean fluorescence intensity for *gstD1-GFP* expression in the PCs of flies reared on either ND or HSD. The dots represent the number of PCs analyzed for each genotype. **(I)** Feeding on HSD leads to an increase in the levels of *upd3* expression in the PCs of flies of all the genotypes mentioned. The transcript levels are normalized to that of the constitutive ribosomal gene *rp49*. **(J)** Increase in the levels of *upd3* expression in the PCs of flies of the genotypes mentioned when reared on HSD. The transcript levels are normalized to that of the constitutive ribosomal gene *rp49*. Genotypes are as mentioned. Data are represented as mean ± SD. P values (ns ≥ 0.05, **P < 0.01, ***P < 1 × 10⁻³) were obtained by unpaired Student's *t* test (two-tailed) with Welch's correction (H), by one-way ANOVA with Tukey's multiple-comparison test (C, D, and E), and by two-way ANOVA with Tukey's multiple-comparison test (B, I, and J). Scale bars, 100 µm (A) and 50 µm (F and G).

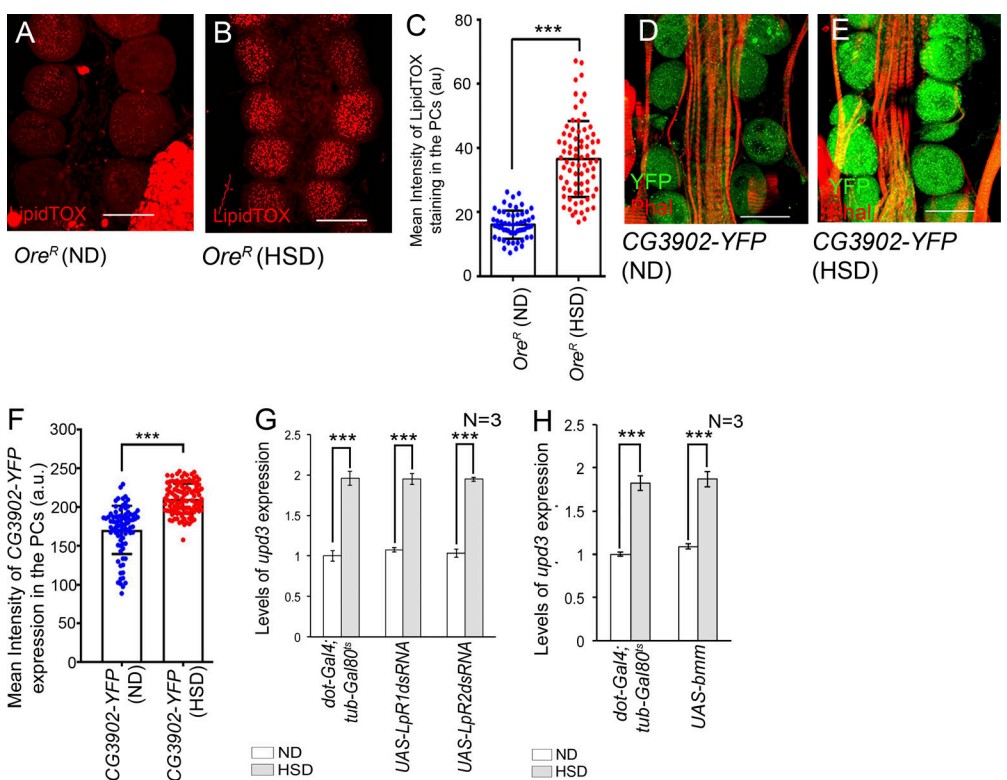

**Figure S3. The PCs of HSD-fed flies exhibit altered lipid metabolism. (A and B)** Increase in the levels of lipid accumulation (LipidTOX staining; red) in the PCs of HSD-fed flies (B) as compared with those reared on ND (A). **(C)** Quantification of the mean fluorescence intensity for LipidTOX staining in the PCs flies reared on either ND or HSD. The dots represent the number of PCs analyzed for each genotype. **(D and E)** Increase in the reporter YFP expression for *CG3902* in the PCs of HSD-fed flies (E) as compared to those fed on ND (D). Phalloidin (red) marks the cardiac tube and the alary muscles. **(F)** Quantification of the mean fluorescence intensity for *CG3902-YFP* expression in the PCs of flies reared on either ND or HSD. The dots represent the number of PCs analyzed for each genotype. **(G)** Increase in the levels of *upd3* expression in the PCs of flies of the genotypes mentioned when reared on HSD. The transcript levels are normalized to that of the constitutive ribosomal gene *rp49*. **(H)** Feeding on HSD leads to an increase in the levels of *upd3* expression in the PCs of flies of the genotypes mentioned. The transcript levels are normalized to that of the constitutive ribosomal gene *rp49*. Genotypes are as mentioned. Data are represented as mean ± SD. P values (***P < 1 × 10$^{-3}$) were obtained by unpaired Student's *t* test (two-tailed) with Welch's correction (C and F) and by two-way ANOVA with Tukey's multiple-comparison test (G and H). Scale bars, 50 µm.

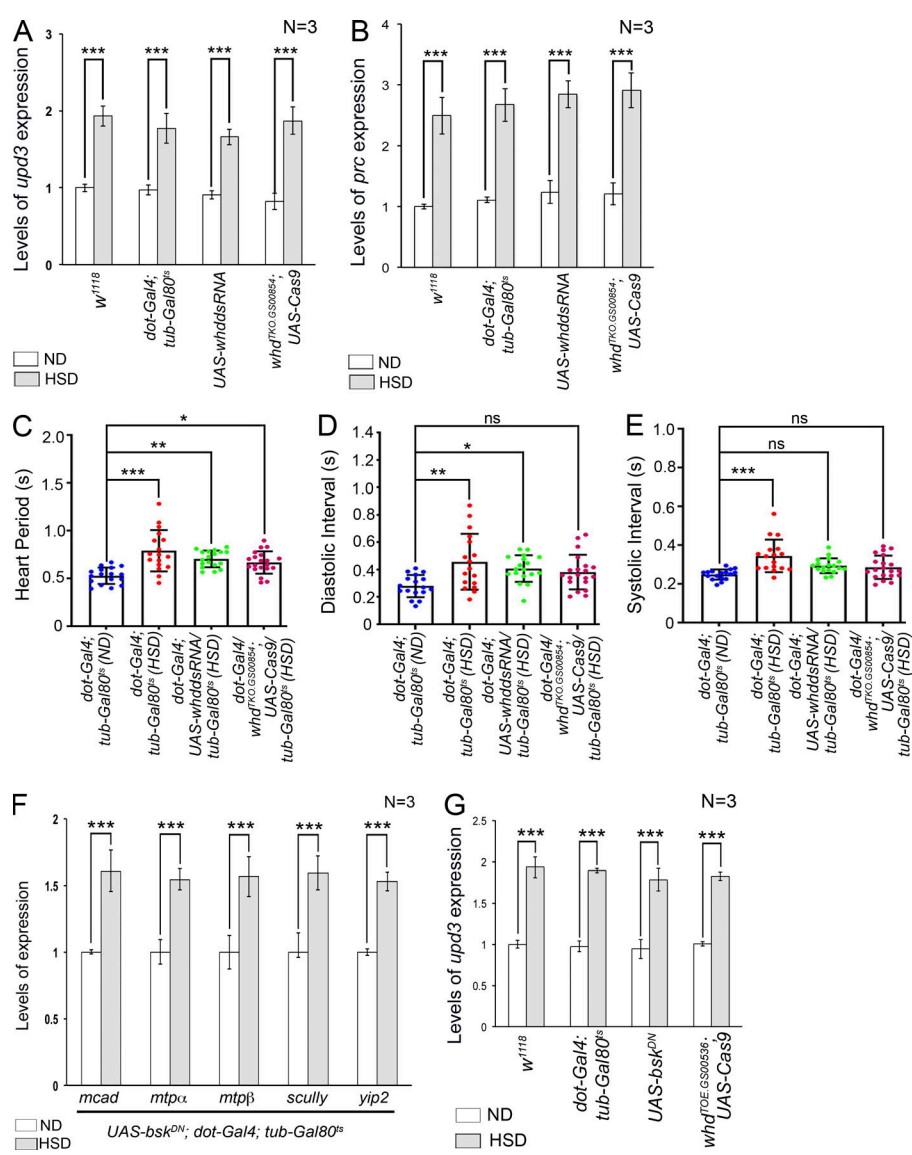

**Figure S4. Attenuation of FAO in the PCs rescue the elevated levels of *upd3* and *prc* expressions as well as abnormal cardiac function associated with high dietary sugar. (A)** Feeding on HSD led to an increase in the levels of *upd3* expression in the PCs of flies of all the genotypes mentioned. The transcript levels are normalized to that of the constitutive ribosomal gene *rp49*. **(B)** Increase in the levels of *prc* expression in fat cells of the flies of the genotypes mentioned when reared on HSD. The transcript levels are normalized to that of the constitutive ribosomal gene *rp49*. **(C–E)** Changes in heart period (C), diastolic interval (D), and systolic interval (E) upon either knocking down or knocking out *whd* in the PCs of HSD-fed flies. The dots represent the samples analyzed for each genotype. **(F)** Levels of expression of different enzymes involved in FAO in the PCs of ND- and HSD-fed flies upon attenuating JNK pathway in the PCs. The transcript levels are normalized to that of the constitutive ribosomal gene *rp49*. **(G)** Increase in the levels of *upd3* expression in the PCs of flies of the genotypes mentioned when reared on HSD. The transcript levels are normalized to that of the constitutive ribosomal gene *rp49*. Genotypes are as mentioned. Data are represented as mean ± SD. P values (ns ≥ 0.05, *P < 0.05, **P < 0.01, ***P < 1 × 10$^{-3}$) were obtained by unpaired Student's *t* test (two-tailed) with Welch's correction (F), by one-way ANOVA with Tukey's multiple comparison test (C, D, and E), and by two-way ANOVA with Tukey's multiple-comparison test (A, B, and G).

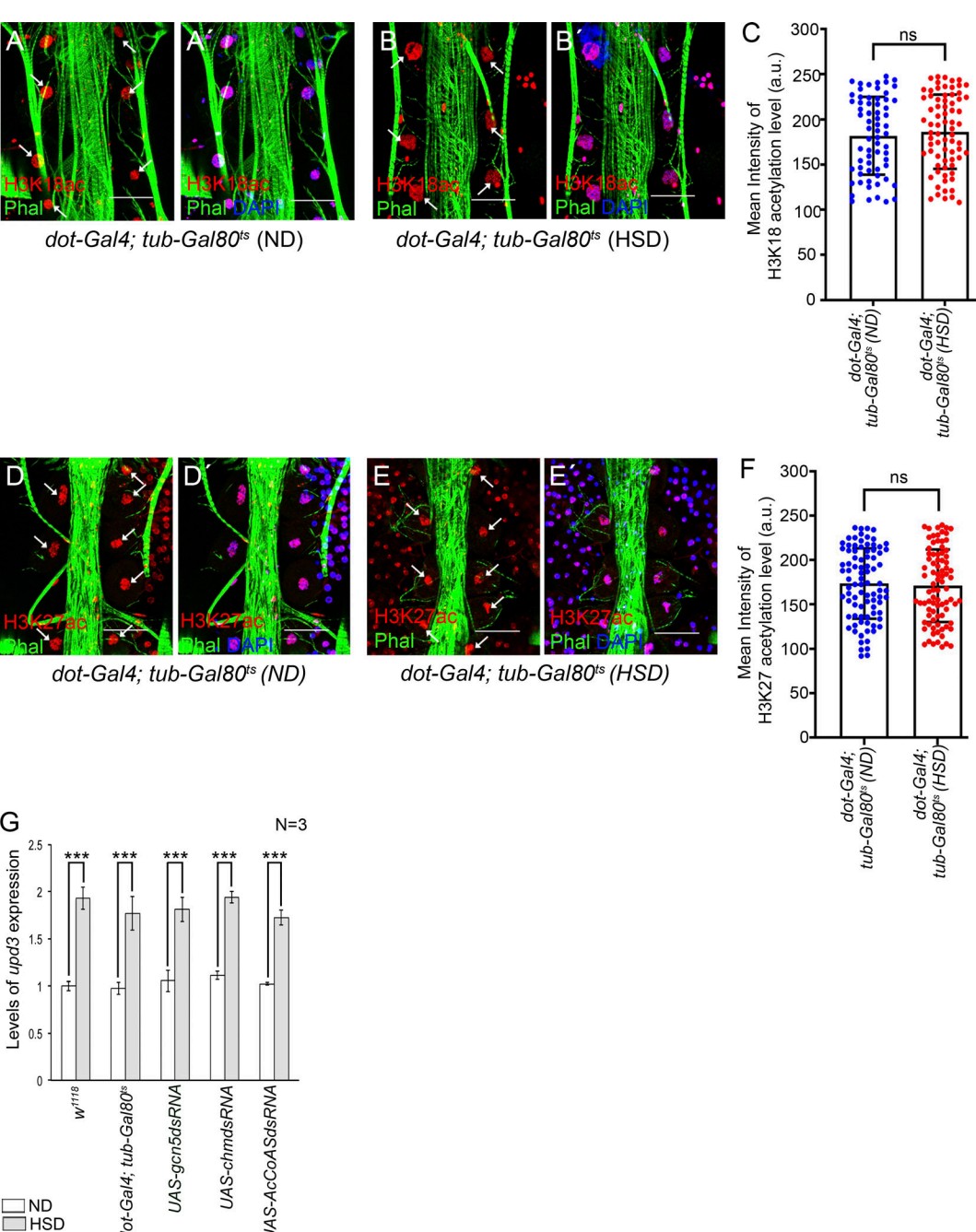

Figure S5.   **High dietary sugar does not impact the levels of H3K18ac and H3K27ac in the PCs. (A–B')** Compared with that observed in the PCs of ND-fed flies (A and A'), the levels of H3K18ac (red) remain unaltered in the PCs of HSD-fed flies (B and B'). Arrows mark the PCs in A and B. Phalloidin (green) marks the cardiac tube and the alary muscles. DAPI marks the nuclei in A' and B´. **(C)** Quantification of the mean fluorescence intensity for H3K18ac in the PCs of ND- and HSD-fed flies. The dots represent the number of PCs analyzed for each genotype. **(D–E')** Compared with that observed in the PCs of ND-fed flies (D and D'), the levels of H3K27ac (red) remain unaltered in the PCs of HSD-fed flies (E and E'). Arrows mark the PCs in D and E. Phalloidin (green) marks the cardiac tube and the alary muscles. DAPI marks the nuclei in D' and E'. **(F)** Quantification of the mean fluorescence intensity for H3K27ac in the PCs of ND- and HSD-fed flies. The dots represent the number of PCs analyzed for each genotype. **(G)** Feeding on HSD led to an increase in the levels of *upd3* expression in the PCs of the flies of all the genotypes mentioned. The transcript levels are normalized to that of the constitutive ribosomal gene *rp49*. Genotypes are as mentioned. Data are represented as mean ± SD. P values (ns ≥ 0.05, ***P < 1 × 10⁻³) were obtained by unpaired Student's *t* test (two-tailed) with Welch's correction (C and F), by two-way ANOVA with Tukey's multiple-comparison test (G). Scale bars, 50 µm.

Video 1.   **Beating second chamber of the cardiac tube (heart) of adult *Drosophila* (*Ore^R*) reared on ND.** Video was captured under bright-field using a QImaging optiMOS scMOS camera R696 M-16-C with a 20× water immersion lens (Carl Zeiss). The neighboring pericardial cells and fat tissue were removed during sample preparation to clearly access the cardiac tube for video recording. The video was captured at 100 fps for a duration of 10 s. Playback speed is 100 fps. This video is related to Fig. 1 H.

Video 2.   **Beating second chamber of the cardiac tube (heart) of adult *Drosophila* (*Ore^R*) reared on HSD.** Video was captured under bright-field using a QImaging optiMOS scMOS camera R696 M-16-C with a 20× water immersion lens (Carl Zeiss). The neighboring pericardial cells and fat tissue were removed during sample preparation to clearly access the cardiac tube for video recording. The video was captured at 100 fps for a duration of 10 s. Playback speed is 100 fps. This video is related to Fig. 1 H.

Video 3.   **Beating second chamber of the cardiac tube (heart) of adult *Drosophila* (*dot-Gal4; tub-Gal80^ts*) reared on ND.** Video was captured under bright-field using a QImaging optiMOS scMOS camera R696 M-16-C with a 20× water immersion lens (Carl Zeiss). The neighboring PCs and fat tissue were removed during sample preparation to clearly access the cardiac tube for video recording. The video was captured at 100 fps for a duration of 10 s. Playback speed is 100 fps. This video is related to Fig. 2 J.

Video 4.   **Beating second chamber of the cardiac tube (heart) of adult *Drosophila* (*dot-Gal4; tub-Gal80^ts*) reared on HSD.** Video was captured under bright-field using a QImaging optiMOS scMOS camera R696 M-16-C with a 20× water immersion lens (Carl Zeiss). The neighboring PCs and fat tissue were removed during sample preparation to clearly access the cardiac tube for video recording. The video was captured at 100 fps for a duration of 10 s. Playback speed is 100 fps. This video is related to Fig. 2 J.

Video 5.   **Beating second chamber of the cardiac tube (heart) of adult *Drosophila* (*dot-Gal4; tub-Gal80^ts/UAS-upd3dsRNA*) reared on HSD.** Video was captured under bright-field using a QImaging optiMOS scMOS camera R696 M-16-C with a 20× water immersion lens (Carl Zeiss). The neighboring PCs and fat tissue were removed during sample preparation to clearly access the cardiac tube for video recording. The video was captured at 100 fps for a duration of 10 s. Playback speed is 100 fps. This video is related to Fig. 2 J.

Video 6.   **Beating second chamber of the cardiac tube (heart) of adult *Drosophila* (*dot-Gal4; tub-Gal80^ts*) reared on ND.** Video was captured under bright-field using a QImaging optiMOS scMOS camera R696 M-16-C with a 20× water immersion lens (Carl Zeiss). The neighboring PCs and fat tissue were removed during sample preparation to clearly access the cardiac tube for video recording. The video was captured at 100 fps for a duration of 10 s. Playback speed is 100 fps. This video is related to Fig. 7 I.

Video 7.   **Beating second chamber of the cardiac tube (heart) of adult *Drosophila* (*dot-Gal4; tub-Gal80^ts*) reared on HSD.** Video was captured under bright-field using a QImaging optiMOS scMOS camera R696 M-16-C with a 20× water immersion lens (Carl Zeiss). The neighboring PCs and fat tissue were removed during sample preparation to clearly access the cardiac tube for video recording. The video was captured at 100 fps for a duration of 10 s. Playback speed is 100 fps. This video is related to Fig. 7 I.

Video 8.   **Beating second chamber of the cardiac tube (heart) of adult *Drosophila* (*dot-Gal4; UAS-whddsRNA/tub-Gal80^ts*) reared on HSD.** Video was captured under bright-field using a QImaging optiMOS scMOS camera R696 M-16-C with a 20× water immersion lens (Carl Zeiss). The neighboring pericardial cells and fat tissue were removed during sample preparation to clearly access the cardiac tube for video recording. The video was captured at 100 fps for a duration of 10 s. Playback speed is 100 fps. This video is related to Fig. 7 I.

Video 9.   **Beating second chamber of the cardiac tube (heart) of adult *Drosophila* (*dot-Gal4/whd^{TKO.GS00854}; UAS-Cas9/tub-Gal80^ts*) reared on HSD.** Video was captured under bright-field using a QImaging optiMOS scMOS camera R696 M-16-C with a 20× water immersion lens (Carl Zeiss). The neighboring PCs and fat tissue were removed during sample preparation to clearly access the cardiac tube for video recording. The video was captured at 100 fps for a duration of 10 s. Playback speed is 100 fps. This video is related to Fig. 7 I.

**Provided online are two tables. Table S1 shows the details of the fly stocks used. Table S2 lists primers used.**

