## [Peer Review File · The Journal of Cell Biology]

High Sugar Diet-Induced Fatty Acid Oxidation Potentiates Cytokine-dependent cardiac ECM remodeling

Jayati Gera, Dheeraj Kumar, Gunjan Chauhan, Adarsh Choudhary, Lavi Rani, Lolitika Mandal, and Sudip Mandal

Corresponding Author(s): Sudip Mandal, Indian Institute of Science Education and Research Mohali

Review Timeline:

Submission Date:	2023-06-21
Editorial Decision:	2023-09-05
Revision Received:	2024-03-09
Editorial Decision:	2024-05-14
Revision Received:	2024-06-05

Monitoring Editor: Mark Peifer

Scientific Editor: Andrea Marat

Transaction Report:

DOI: <https://doi.org/10.1083/jcb.202306087>

September 5, 2023

Re: JCB manuscript #202306087

Dr. Sudip Mandal
Indian Institute of Science Education and Research Mohali
Department of Biological Sciences, Academic Block I IISER Mohali Sector 81, Knowledge City
Mohali
Mohali, Punjab 140306
India

Dear Dr. Mandal,

Thank you for submitting your manuscript entitled "High Sugar Diet-Induced Fatty Acid Oxidation Potentiates Cytokine-dependent cardiac ECM remodeling". The manuscript was assessed by expert reviewers, whose comments are appended to this letter. We invite you to submit a revision if you can address the reviewers' key concerns, as outlined here.

As you will see, both reviewers found the topic of interest, but had concerns over the strength of the data supporting some of your conclusions. We'd particularly note:

- 1) issues with quantification, concerning image saturation
- 2) Concerns about the qPCR data
- 3) Issues regarding support for your explicit model that HSD induces a FAO-JNK-Jak/STAT pathway.

These issues would need to be addressed experimentally, and doing so would likely require expansion of the work to an Article rather than a Report.

GENERAL GUIDELINES:

Text limits: Character count for an Report is < 40,000, not including spaces. Count includes title page, abstract, introduction, results, discussion, and acknowledgments. Count does not include materials and methods, figure legends, references, tables, or supplemental legends.

Figures: Reports may have up to 10 main text figures. Figures must be prepared according to the policies outlined in our Instructions to Authors, under Data Presentation, <https://jcb.rupress.org/site/misc/ifora.xhtml>. All figures in accepted manuscripts will be screened prior to publication.

*****IMPORTANT:** It is JCB policy that if requested, original data images must be made available. Failure to provide original images upon request will result in unavoidable delays in publication. Please ensure that you have access to all original microscopy and blot data images before submitting your revision. ***

Supplemental information: There are strict limits on the allowable amount of supplemental data. Reports may have up to 5 supplemental figures. Up to 10 supplemental videos or flash animations are allowed. A summary of all supplemental material should appear at the end of the Materials and methods section.

Please note that JCB now requires authors to submit Source Data used to generate figures containing gels and Western blots with all revised manuscripts. This Source Data consists of fully uncropped and unprocessed images for each gel/blot displayed in the main and supplemental figures. Since your paper includes cropped gel and/or blot images, please be sure to provide one Source Data file for each figure that contains gels and/or blots along with your revised manuscript files. File names for Source Data figures should be alphanumeric without any spaces or special characters (i.e., SourceDataF#, where F# refers to the associated main figure number or SourceDataFS# for those associated with Supplementary figures). The lanes of the gels/blots should be labeled as they are in the associated figure, the place where cropping was applied should be marked (with a box), and molecular weight/size standards should be labeled wherever possible.

The typical timeframe for revisions is three to four months. While most universities and institutes have reopened labs and allowed researchers to begin working at nearly pre-pandemic levels, we at JCB realize that the lingering effects of the COVID-19 pandemic may still be impacting some aspects of your work, including the acquisition of equipment and reagents. Therefore, if you anticipate any difficulties in meeting this aforementioned revision time limit, please contact us and we can work with you to find an appropriate time frame for resubmission. Please note that papers are generally considered through only one revision cycle, so any revised manuscript will likely be either accepted or rejected.

Thank you for this interesting contribution to Journal of Cell Biology. You can contact us at the journal office with any questions, cellbio@rockefeller.edu or call (212) 327-8588.

Sincerely,

Mark Peifer
Monitoring Editor

Andrea L. Marat
Senior Scientific Editor

Journal of Cell Biology

Reviewer #1 (Comments to the Authors (Required)):

The manuscript by Gera et al. uses the *Drosophila* heart tube and associated pericardial cells to examine the effects of high sugar diet on cardiac fibrosis. This study provides a very interesting follow-up to their recent publication describing physiological ROS activation of JNK/p38 to trigger upd3 release from pericardial cells, which in turn activates JAK-STAT in fat cells, leading to ECM protein production and deposition around the heart tube. The current study makes a compelling argument that high sugar diet (a known contributor to cardiac fibrosis and dysfunction) activates JNK signaling in a ROS-independent manner, and instead leads to altered glycolysis and increased fatty acid oxidation (FAO). They go on to suggest that the increased FAO in these cells promotes histone acetylation, which affects expression of Upd3. The manuscript is well-written, the figures are generally clear and well-organized, and the major analyses are thoroughly quantified. Given that the novel findings in this study provide significant insight into the complex relationships between diet, metabolism, cell signaling, and ECM protein expression, I feel this article is appropriate for publication as a Report in JCB. I have several comments below that I hope will clarify a few aspects of this study.

1. Na et al., 2013 PLoS Genetics and Na et al., 2015 Cell Reports identified the hexosamine flux pathway as an important mediator of high sugar diets in fly pericardial cells. They proposed that changes in hexosamine activity lead to changes in OGT, which alters glycosylation of nuclear proteins, with subsequent changes in gene expression and heart function (including mediating arrhythmia resulting from high sugar diet). In the current study, the authors also identify altered hexosamine activity in high sugar diet animals (in the context of upd3 regulation and arrhythmia), but suggest the effects are mediated by alterations in fatty acid oxidation and histone acetylation. It would be good if the authors could add some discussion comparing these findings and conclusions.
2. The observation that high sugar diet leads to reduced uptake of 2-NBDG in fly nephrocytes has been previously reported by Na et al., Cell Reports, 2015. It seems reasonable that this paper should be cited alongside this observation.
3. The model proposed suggests that increased FAO leads to increased levels of Acetyl-CoA, which promotes histone acetylation. Can the authors speculate as to why the increased acetylation seems specific to H3K9 (no increased acetylation at H3K18 or H3K27 was observed).
4. In Figure 4R, the authors compare the effects of overexpressing whd when JNK signaling is inhibited (via bskDN). If it is acceptable to directly compare the experiments, it would be helpful to include the data in this graph showing the effect of bskDN alone on upd3 expression in HSD (instead of having to refer back to Figure 2K). Also, what is the effect of overexpression of whd alone on upd3 expression in the absence of JNK inhibition (bskDN), or just in normal diet fed flies? Is it sufficient to induce upd3 expression?
5. Pg.7, line 151. The writing indicates there was an increase in upd3 transcripts in the pericardial cells of HSD flies, however, the methods section suggests that the RT-qPCR analyses was performed on heart tubes, which contain additional cells, not just the pericardial cells. Although it is certainly likely that the increase in upd3 RNA is coming from the pericardial cells, the

language should acknowledge that the analysis was not strictly performed on these cells alone.

6. Figure 4. The images in panels F and G are supposed to demonstrate the rescue of Prc levels in HSD. While they do appear different from E, the control image in D appears noticeably lower than F and G. The quantification in panel H is convincing but suggests that the whd knockouts should be very similar to ND controls, on average. If so, can other images be shown that better represent the Prc protein levels in these different backgrounds.

7. Based on image quantification, it appears that the signal from some of the images was saturated during image acquisition (for example, Figure 2O, Figure 4O, and Figure 5H). While this does not change the conclusions made from these data, it may be worth mentioning in the text that this saturation may result in underestimating the actual increases in some protein levels/reporters.

8. Pg.3, line 43. "On the flip side..." is too informal and not necessary. I suggest removing this from the sentence.

Reviewer #2 (Comments to the Authors (Required)):

Mandal and colleagues examine how diet-induced obesogenic remodeling increases cardiac ECM fibrosis using a fruit model in this study. In prior work by the same group, they identified an interorgan communication network between pericardial cells (PF) and the fat body (FB) that results in cardiac ECM accumulation. According to the prior model, under physiological conditions, ROS production from PC activates JNK signaling; this results in the expression of the JAK/STAT pathway ligand Upd3. PC-derived Upd3 activates pericardin (collagen IV-like molecule) equivalent in the FB, which is then secreted and decorates the heart tube as ECM. This study attempts to understand how prc deposits in the heart tube increase. They present a model that pericardin (prc) accumulation in the heart tube during a high sugar diet (HSD) state differs from normal diet (ND). They conclude that the genetic mechanism they present reveals an unexpected role for FAO in ECM remodeling during diabetic fibrosis. Overall, this is an interesting idea. The study leverages the sophisticated genetic toolset in the Drosophila, in combination with the HSD-dietary paradigm, and is the strength of the work.

While this reviewer is convinced that it is not the ROS-dependent circuit, and in the HSD regime different one is utilized, which is likely FAO based. However, data presented regarding the model's following aspects are tenuous: i) due to impaired glycolysis on HSD PCs specifically utilizing Fatty acid as a lipid source; ii) FAO upregulates HDAC-dependent transcription. The technical concerns (discussed below) with imaging datasets and qPCR sample prep significantly diminish enthusiasm. Also, the manuscript was quite challenging to read. There are many threads, but only some of which are well-explored. The authors might want to simplify and deepen the message of their primary model that HSD induces an FAO-JNK-Jak/STAT pathway, as opposed to a ROS-dependent pathway in ND, and to increase the technical rigor of their study, and perhaps explore the HDAC-dependent pathway link in a future work.

1. The primary concern is the image-based analysis of what appears to be saturated confocal images from the presentation of data. A few specific examples are listed below to highlight the problem, but the problem is more extensive than the few examples listed below.

a) Fig. 2H, I, L, M, N: The difference in TRE-dsred intensity is marginal. The red channel appears saturated in the representative images.

b) Figure 4K-N- again, the same problem as above with TRE-desREd imaging.

c) Figure 5F-G'- the anti-H3K9 staining looks odd in F and F' as there are many more nuclei, and how do they know which is the pericardial cell?

The authors might want a look-up table (LUT) to indicate that they are in the linear range and an unmerged grey-scale view of pericardial cells. The image quantification methodology is also unclear. They say mean intensity is used, but is it a 3D-volumetric image or a maximum intensity projection? The data are quantified from pericardial cells, but how many are present in a Field of view in an animal? The data would be more reliable if they indicated how many animals were examined under a specific condition instead of using single pericardial cells from an animal to interpret results.

2. The authors rely extensively on qPCR results from cell types- pericardial cells, fat body, and tissues that are hard to remove from cuticle and muscle. But the authors' interpretation based on qPCR data from a mixed pool of tissue to make a complex argument about inter-organ signals from PC FB Cardiac ECM is puzzling. They could consider FACS sorting the cells or performing snRNAseq to make such strong claims (PMID: 34612794; 32396065; 35239393).

3. The authors do provide evidence that PC cells have low glycolytic activity. But they conjecture that reduced glycolytic activity would increase FAO without presenting evidence that free fatty acids (FFA) are upregulated on HSD. They show that LDs (neutral lipids) are elevated in PC (Figure 3J-L and Figure S2 D-F). Hence, PCs may use lipolysis and not just FAO to generate energy. Another alternative is that increased neutral lipids could reduce ROS, given their antioxidant-like properties (PMID: 34423827). It would be important for the authors to distinguish between these possibilities.

Some experiments are:

-Could the authors examine whether FFAs increase using lipidomics (PMID: 34423827) to assess whether FFAs increase in the PCs/heart?

- Express ATGL/Bmm lipase in the PC on HSD and see whether the ROS level increases? Or does increase lipolysis generate more FFA for FOA?

4. The author's claim that FAO activates HDAC-dependent transcription is not well supported by results presented in Figure 5 C-D', F-G' and the qPCR data in 5C (due to technical concerns raised in #1 and #2 points above). Further evidence is required to bolster this aspect of the study.

Response to Reviewers' Comments
(JCB Manuscript # 202306087)

Reviewer 1:

The manuscript by Gera et al. uses the *Drosophila* heart tube and associated pericardial cells to examine the effects of high sugar diet on cardiac fibrosis. This study provides a very interesting follow-up to their recent publication describing physiological ROS activation of JNK/p38 to trigger upd3 release from pericardial cells, which in turn activates JAK-STAT in fat cells, leading to ECM protein production and deposition around the heart tube. The current study makes a compelling argument that high sugar diet (a known contributor to cardiac fibrosis and dysfunction) activates JNK signaling in a ROS-independent manner, and instead leads to altered glycolysis and increased fatty acid oxidation (FAO). They go on to suggest that the increased FAO in these cells promotes histone acetylation, which affects expression of Upd3. The manuscript is well-written, the figures are generally clear and well-organized, and the major analyses are thoroughly quantified. Given that the novel findings in this study provide significant insight into the complex relationships between diet, metabolism, cell signaling, and ECM protein expression, I feel this article is appropriate for publication as a Report in JCB.

Our sincere thanks to the reviewer for nicely summarizing our work and finding it novel, exciting, and appropriate for publication in JCB.

I have several comments below that I hope will clarify a few aspects of this study.

1. Na et al., 2013 PLoS Genetics and Na et al., 2015 Cell Reports identified the hexosamine flux pathway as an important mediator of high sugar diets in fly pericardial cells. They proposed that changes in hexosamine activity lead to changes in OGT, which alters glycosylation of nuclear proteins, with subsequent changes in gene expression and heart function (including mediating arrhythmia resulting from high sugar diet). In the current study, the authors also identify altered hexosamine activity in high sugar diet animals (in the context of upd3 regulation and arrhythmia), but suggest the effects are mediated by alterations in fatty acid oxidation and histone acetylation. It would be good if the authors could add some discussion comparing these findings and conclusions.

We thank the reviewer for raising this aspect!

Recent studies involving fly and mouse models have attributed flux through the hexosamine biosynthetic pathway to diet-induced cardiac dysfunction. It has been documented that increasing hexosamine flux phenocopies HSD-mediated heart dysfunction and genetic attenuation of the hexosamine biosynthetic pathway, specifically in the adult heart, leads to improved heart function of HSD-fed flies (Na et al., 2013 PLoS Genetics). Though the underlying mechanism is not evident, at least it is clear that the restoration in cardiac function is not associated with any detectable alteration in the elevated levels of Pericardin accumulation observed in HSD-fed flies (Jinabo Na et al.; PLoS Genetics, 2013). A separate study provides evidence for the

involvement of the high dietary sugar-induced hexosamine pathway in the etiology of PC dysfunction in adult flies for its role in epigenetic modification-induced altered gene expression (Na et al., 2015 Cell Reports). From that perspective, our study's outcome provides evidence for an alternate epigenetic mechanism by which high dietary sugar modulates gene expression in the PCs and braids, with the previous observations providing a better understanding of high dietary sugar-induced diabetic nephropathy and cardiac dysfunction. We conjecture that an increased rate of FAO performs two distinct roles in regulating gene expression in the PCs of HSD-fed flies. On the one hand, it promotes hexosamine pathway-dependent O-GlycNAcylation of the Polyhomeotic subunit of PRC1 by supplying Acetyl CoA, essential for the hexosamine biosynthetic pathway; on the other hand, Acetyl CoA is utilized for histone acetylation to modulate *upd3* expression. While the first process exacerbates the nephrocytic function of the PCs, the second one aggravates cardiac dysfunction by augmenting excessive accumulation of Prc around the cardiac tube. Furthermore, our results, in conjunction with the previous findings, elucidate that activation of the JNK-FAO-H3K9Ac axis in the PCs of HSD-fed flies enhances cardiac dysfunction by increased Prc accumulation around the heart, whereas escalated hexosamine flux acts on other specific, yet unknown, components of the heart to induce cardiac dysfunction of HSD-fed flies. This might be why the increase in Prc accumulation around the cardiac tube of HSD-fed flies remained unaltered even after heart-specific attenuation of the hexosamine pathway.

We have included this in the Discussion section of the revised manuscript. **Please refer to paragraph 2 of the Discussion.**

Comparing our results with these previous findings has indeed increased the quality of this manuscript.

2. The observation that high sugar diet leads to reduced uptake of 2-NBDG in fly nephrocytes has been previously reported by Na et al., Cell Reports, 2015. It seems reasonable that this paper should be cited alongside this observation.

We are sorry for not referring this study alongside our observation! We have now referred this paper in the revised version of the manuscript. **Kindly refer to page...of the revised manuscript.**

3. The model proposed suggests that increased FAO leads to increased levels of Acetyl-CoA, which promotes histone acetylation. Can the authors speculate as to why the increased acetylation seems specific to H3K9 (no increased acetylation at H3K18 or H3K27 was observed).

Increased levels of H3K9ac, but not H3K18ac or H3K27ac, suggest that high dietary sugar does not lead to increased acetylation of all lysine residues of histones. Instead, site-specific histone acetylation generates a unique histone acetylation signature responsible for modulating gene expression in the PCs. Of note, several studies in the recent past have documented the dietary influence on site-specific histone acetylation in different organs. For instance, high-fat feeding in mice has been associated with reduced H3K18ac and H3K23ac levels in white adipose tissue and the pancreas (Carrer A. et al., 2017, J. Biol. Chem), while hepatic gene transcription regulation is associated with changes in H3K27ac levels (Siersbæk M. et al., 2017, Scientific

Reports). Likewise, in type 2 diabetic mice, progressive glomerulosclerosis is associated with renal histone H3K9 and H3K23 acetylation (Sufyan G Sayyed et al., 2010, Nephrology Dialysis Transplantation). Therefore, our findings provide another example of

We have included this in the Discussion of the revised manuscript. **Please refer to the last part of the first paragraph of the Discussion.**

4. In Figure 4R, the authors compare the effects of overexpressing *whd* when JNK signaling is inhibited (via *bskDN*). If it is acceptable to directly compare the experiments, it would be helpful to include the data in this graph showing the effect of *bskDN* alone on *upd3* expression in HSD (instead of having to refer back to Figure 2K).

Also, what is the effect of overexpression of *whd* alone on *upd3* expression in the absence of JNK inhibition (*bskDN*), or just in normal diet fed flies? Is it sufficient to induce *upd3* expression?

We are sorry for the inconvenience caused! In the revised manuscript, we have included the effects of *bskDN* expression on *upd3* expression in both ND and HSD conditions in the same figure. As suggested by the reviewer, we also checked for the effects of *whd* overexpression (in the absence of JNK inhibition) on *upd3* expression in both ND and HSD conditions. We included the result in the same figure.

Under normal dietary conditions, overexpression of *whd* in the PCs led to a remarkable increase in *upd3* transcripts compared to control. However, overexpression of *whd* in the PCs of HSD-fed flies resulted in a modest, but not statistically significant, increase in *upd3* expression compared to that observed in HSD-fed flies. There are two possible reasons for not having further elevation in *upd3* expression upon upregulating *whd* in the PCs of HSD flies: 1) Even though overexpression of *whd* could, in principle, cause increased mobilization of free fatty acids into FAO, it might not lead to any further increase in FAO as these cells already have high levels of *whd* expression and increased FAO due to high dietary sugar. 2) Since elevated activation of the *whd*-FAO axis upregulates *upd3* expression in the PCS of HSD-fed flies by histone acetylation, a saturation of histone acetylation sites might be a limiting factor for further increase.

We have included these results in the revised manuscript. Please refer to Figure 8G and on page...

5. Pg.7, line 151. The writing indicates there was an increase in *upd3* transcripts in the pericardial cells of HSD flies, however, the methods section suggests that the RT-qPCR analyses was performed on heart tubes, which contain additional cells, not just the pericardial cells. Although it is certainly likely that the increase in *upd3* RNA is coming from the pericardial cells, the language should acknowledge that the analysis was not strictly performed on these cells alone.

We agree with the reviewer that RNA was isolated from the tissue, including PCs and heart tubes, to assess the level of *upd3* transcripts in the PCs. Technically, it is next to impossible to separate the PCs from the cardiac tubes. Given their relatively larger size, the PCs get easily disrupted while preparing single-cell suspensions. Due to

these technical limitations, we resorted to isolating RNAs from the PCs and the cardiac tissue.

Moreover, in most cases, we checked for the abundance of *upd3* transcripts in the RNA isolated from the PCs and the cardiac tubes. Since *upd3* expresses only in the PCs but not in the cardiac tube (Figure 2A-B of the revised manuscript; Gera et al., 2022, Sci Adv), the changes in *upd3* expression reflect the changes in the PCs.

However, this argument may not hold well when we try to assess the changes in expression levels of genes associated with lipid metabolism (**these results have been included in the revised manuscript**) in the PCs of HSD-fed flies. Therefore, in such cases, we have explicitly mentioned that the changes in the transcript levels were observed in the RNA isolated from the PCs and the cardiac tube. **Please refer to the following sections of the revised manuscript:**

6. Figure 4. The images in panels F and G are supposed to demonstrate the rescue of Prc levels in HSD. While they do appear different from E, the control image in D appears noticeably lower than F and G. The quantification in panel H is convincing but suggests that the whd knockouts should be very similar to ND controls, on average. If so, can other images be shown that better represent the Prc protein levels in these different backgrounds.

Our sincere thanks to the reviewer for pointing this out. We have now replaced the control image for Prc incorporation around the second cardiac chamber with a better representation. **Please refer to Figure 7D in the revised manuscript.**

7. Based on image quantification, it appears that the signal from some of the images was saturated during image acquisition (for example, Figure 2O, Figure 4O, and Figure 5H). While this does not change the conclusions made from these data, it may be worth mentioning in the text that this saturation may result in underestimating the actual increases in some protein levels/reporters.

Thanks for pointing out this mistake! We understand and appreciate the Reviewer's concern and admit that the signal in some of the images related to TRE-DsRed expression was saturated during image acquisition (Figures 2O and 4O of the original manuscript). Our sincere apologies for this inadvertent mistake!

To rectify this error, we again performed the experiments (analyzing TRE-DsRed expression in the PCs of all genotypes). During image acquisition, care was taken to avoid any saturation issues.

In the revised manuscript, we provided new images for TRE-dsRed expressions in all genotypic backgrounds. We have also included the unmerged grey-scale view of pericardial cells in all cases. **Please refer to Figures 3H-I', 3L-N', and 8A-D' of the revised manuscript.**

Quantitative analyses of PC-specific TRE-dsRed expression were done on the new set of images, and the **results are presented in Figures 3J, 3O, and 8E of the revised manuscript.**

Likewise, we again performed immunostaining of PCs with antibodies against H3K9ac for all the genotypes. Imaging of the new samples was done without any

saturation of fluorescence signal. **The revised manuscript shows the new images in Figures 9C-F" and 9H-J".**

Quantitative analyses of the levels of H3K9ac were done on the new set of images, and the **results are presented in Figures 9G and 9K of the revised manuscript.**

Though these new results do not change the conclusions made in the original version of the manuscript, they have surely rectified the technical error associated with image acquisition in the earlier version.

8. Pg.3, line 43. "On the flip side..." is too informal and not necessary. I suggest removing this from the sentence.

Our sincere apologies for using this informal term! We have removed this from the sentence.

Reviewer 2:

Mandal and colleagues examine how diet-induced obesogenic remodeling increases cardiac ECM fibrosis using a fruit model in this study. In prior work by the same group, they identified an interorgan communication network between pericardial cells (PF) and the fat body (FB) that results in cardiac ECM accumulation. According to the prior model, under physiological conditions, ROS production from PC activates JNK signaling; this results in the expression of the JAK/STAT pathway ligand Upd3. PC-derived Upd3 activates pericardin (collagen IV-like molecule) equivalent in the FB, which is then secreted and decorates the heart tube as ECM. This study attempts to understand how prc deposits in the heart tube increase. They present a model that pericardin (prc) accumulation in the heart tube during a high sugar diet (HSD) state differs from normal diet (ND). They conclude that the genetic mechanism they present reveals an unexpected role for FAO in ECM remodeling during diabetic fibrosis. Overall, this is an interesting idea. The study leverages the sophisticated genetic toolset in the *Drosophila*, in combination with the HSD-dietary paradigm, and is the strength of the work.

Our sincere thanks to the reviewer for her/his encouraging comments and appreciation of our work!!

While this reviewer is convinced that it is not the ROS-dependent circuit, and in the HSD regime different one is utilized, which is likely FAO based. However, data presented regarding the model's following aspects are tenuous: i) due to impaired glycolysis on HSD PCs specifically utilizing Fatty acid as a lipid source; ii) FAO upregulates HDAC-dependent transcription. The technical concerns (discussed below) with imaging datasets and qPCR sample prep significantly diminish enthusiasm. Also, the manuscript was quite challenging to read. There are many threads, but only some of which are well-explored. The authors might want to simplify and deepen the message of their primary model that HSD induces an FAO-JNK-Jak/STAT pathway, as opposed to a ROS-dependent pathway in ND, and to increase the

technical rigor of their study, and perhaps explore the HDAC-dependent pathway link in a future work.

We understand and appreciate the Reviewer's two significant concerns. We have specifically addressed both concerns, and our responses are given below.

We are sorry that the Reviewer had difficulty in reading. We have tried our best to improve the manuscript's readability in the revised version. However, we are open to suggestions/inputs that further improve the readability.

1. The primary concern is the image-based analysis of what appears to be saturated confocal images from the presentation of data. A few specific examples are listed below to highlight the problem, but the problem is more extensive than the few examples listed below.
 - a) Fig. 2H, I, L, M, N: The difference in TRE-dsred intensity is marginal. The red channel appears saturated in the representative images.
 - b) Figure 4K-N- again, the same problem as above with TRE-desREd imaging.

Thanks for pointing out this mistake! We understand and appreciate the Reviewer's concern and admit that the signal in some of the images related to TRE-DsRed expression was saturated during image acquisition (Figures 2H, I, L, M, N and 4K-N of the original manuscript). Our sincere apologies for this inadvertent mistake!

To rectify this error, we again performed the experiments (analyzing TRE-DsRed expression in the PCs of all genotypes). During image acquisition, care was taken to avoid any saturation issues.

In the revised manuscript, we provided new images for TRE-dsRed expressions in all genotypic backgrounds. We have also included the unmerged grey-scale view of pericardial cells in all cases. **Please refer to Figures 3H-I', 3L-N', and 8A-D' of the revised manuscript.**

Quantitative analyses of PC-specific TRE-dsRed expression were done on the new set of images, and the **results are presented in Figures 3J, 3O, and 8E of the revised manuscript.**

Likewise, we again performed immunostaining of PCs with antibodies against H3K9ac for all the genotypes. Imaging of the new samples was done without any saturation of fluorescence signal. **The revised manuscript shows the new images in Figures 9C-F" and 9H-J".**

Quantitative analyses of the levels of H3K9ac were done on the new set of images, and the **results are presented in Figures 9G and 9K of the revised manuscript.**

Though these new results do not change the conclusions made in the original version of the manuscript, they have surely rectified the technical error associated with image acquisition in the earlier version.

- c) Figure 5F-G'- the anti-H3K9 staining looks odd in F and F' as there are many more nuclei, and how do they know which is the pericardial cell?

We are sorry for the confusion caused!

While repeating immunostaining of the tissues with antibodies against H3K9ac, in all cases, we counterstained the tissues with an antibody against Prospero that explicitly marks the PCs. The red (H3K9ac) and green (Pros) signals overlap, marking the PCs. In addition, the nuclei of the PCs have been marked with arrows in all images. **Please refer to Figures 9C-C", D-D", E-E", F-F", H-H", I-I", and J-J" of the revised manuscript.**

The authors might want a look-up table (LUT) to indicate that they are in the linear range and an unmerged grey-scale view of pericardial cells. The image quantification methodology is also unclear. They say mean intensity is used, but is it a 3D-volumetric image or a maximum intensity projection? The data are quantified from pericardial cells, but how many are present in a Field of view in an animal? The data would be more reliable if they indicated how many animals were examined under a specific condition instead of using single pericardial cells from an animal to interpret results.

As suggested by the Reviewer, we have provided a grey-scale view of the pericardial cells for TRE-dsRed staining in all genotypic backgrounds in the revised manuscript. **Please refer to Figures 3H, I, L-N, and Figures 8A-D of the revised manuscript.**

We are sorry for not specifying how we obtained the mean expression intensity for the analyzed images. For image quantification, we determined the mean pixel intensity of the desired area of maximum intensity projections of the Z stacks by Image J. We have **mentioned this in the Materials and Methods section of the revised manuscript.**

For image-based quantification of the expression levels in PCs, 50-70 PCs from 12-15 individual flies were analyzed for each genotype/dietary alteration. In particular, the PCs around the cardiac tube's second chamber were analyzed for uniformity.

To determine the intensity of Pericardin accumulation around the second chamber of the cardiac tube, the entire second chamber was marked, and the mean fluorescence intensity (mean grey pixel intensity of the desired area of maximum intensity projections of the Z stacks) was measured using Fiji software. Data were obtained from 20 individual flies to analyze the intensity of Pericardin accumulation around the second chamber of the cardiac tube. **These details are now included in the Materials and Methods section of the revised manuscript.**

We thank the Reviewer for allowing us to clarify these methodological details for the readers.

2. The authors rely extensively on qPCR results from cell types- pericardial cells, fat body, and tissues that are hard to remove from cuticle and muscle. But the authors' interpretation based on qPCR data from a mixed pool of tissue to make a complex argument about inter-organ signals from PC \diamond FB \diamond Cardiac ECM is puzzling. They could consider FACS sorting the cells or performing snRNAseq to make such strong claims (PMID: 34612794; 32396065; 35239393).

We agree with the reviewer that RNA was isolated from the tissue, including PCs and heart tubes, to assess the level of *upd3* transcripts in the PCs. Technically, it is next to impossible to separate the PCs from the cardiac tubes. Given their relatively larger size, the PCs get easily disrupted while preparing single-cell suspensions. Due to these technical limitations, we resorted to isolating RNAs from the PCs and the cardiac tissue.

Moreover, in most cases, we checked for the abundance of *upd3* transcripts in the RNA isolated from the PCs and the cardiac tubes. Since *upd3* expresses only in the PCs but not in the cardiac tube (Figure 2A-B of the revised manuscript; Gera et al., 2022, Sci Adv), the changes in *upd3* expression reflect the changes in the PCs.

However, this argument may not hold well when we try to assess the changes in expression levels of genes associated with lipid metabolism (**these results have been included in the revised manuscript**) in the PCs of HSD-fed flies. Therefore, in such cases, we have explicitly mentioned that the changes in the transcript levels were observed in the RNA isolated from the PCs and the cardiac tube. **Please refer to the following sections of the revised manuscript:**

Regarding the analyses of *prc* expression in the fat cells: For this purpose, RNA was not isolated from a mixed tissue that includes fat cells, cuticle and cuticular muscles. Instead, after dissecting out the abdominal cuticles (with fats, cuticular muscles, attached to them) in Schneider's insect medium on ice, the fat bodies were carefully dissected out and collected in a 1.5ml micro-centrifuge tube with Schneider's insect medium on ice. The samples were then centrifuged for 5 minutes at 5000 RPM (4°C) to settle down the fat cells at the bottom of the tube. Then the samples were homogenized in Schneider's insect medium and the lysate was passed through a 22G syringe (BD No. 309631) 2-3 times. The collected lysate was centrifuged for 5 minutes at 5000 RPM (4°C) to pellet down the cells. The Schneider's insect medium was replaced with TRIzol (Ambion no. 15596018) and RNA was isolated according to the manufacturer's instructions.

Actually, the method followed to isolate the adult fat cells is a modification of the procedure reported in one of the studies (PMID: 34612794) referred by the Reviewer. However, instead of isolation of the fat nuclei (Gupta and Lazzaro 2022; PMID: 34612794), we lysed the cells for RNA extraction. **The detailed procedure has been included in the Materials and Methods section of the revised manuscript.**

Furthermore, from the RNA isolated by this procedure we only checked for the transcript levels of *prc*, that expresses only in the fat cells but not in the abdominal muscles and the cuticle (Fig. 1L, M; Gera et al., 2022). As shown in the figure below *prc* expression () is not seen in the cuticular cells, cardiac tube, PCs and abdominal muscles.

Therefore, we believe that the changes in the levels of *prc* expression are associated with the fat cells.

3. The authors do provide evidence that PC cells have low glycolytic activity. But they conjecture that reduced glycolytic activity would increase FAO without presenting evidence that free fatty acids (FFA) are upregulated on HSD. They show that LDs (neutral lipids) are elevated in PC (Figure 3J-L and Figure S2 D-F). Hence, PCs may use lipolysis and not just FAO to generate energy. Another alternative is that increased neutral lipids could reduce ROS, given their antioxidant-like properties (PMID: 34423827). It would be important for the authors to distinguish between these possibilities.

We sincerely thank the Reviewer for raising these questions!

This prompted us to perform a series of experiments to analyze the status of lipid metabolism in the PCs of HSD-fed flies. The results of those experiments helped us better understand the underlying mechanism and the metabolic connection that leads to increased *upd3* expression in the PCs of HSD-fed flies. The essential findings are as follows:

- a) Quantitative analyses of the transcript levels of the genes encoding enzymes for lipogenesis, which include *Gpat4*, *Agpat4*, and *Dgat2*, revealed their enrichment in the RNA isolated from PCs and cardiac tubes of HSD-fed flies as compared to that of ND-fed flies (**Fig. 5B**). Interestingly, compared to *Gpat4* and *Agpat4*, a robust increase in the levels of *Dgat2* transcripts (which codes for the enzyme involved in the conversion of DAG to TAG) was detected.
- b) Likewise, expression of the genes involved in lipolysis, such as *bmm* and *Hsl*, were also upregulated in HSD-fed conditions (**Fig. 5B**).
- c) A significant reduction in the levels of *Acc* and *Fasn1* expression (genes involved in *de novo* fatty acid synthesis) was observed (**Fig. 5B**).

These results indicate that high dietary sugar induced significant alterations in cellular lipid metabolism. At the same time, the kinetics of both lipogenesis (robust in case of conversion of DAG to TAG) and lipolysis are upregulated, and the generation of fatty acids by *de novo* fatty acid synthesis is compromised. Since the rate of *de novo* fatty acid synthesis depends on glycolytic flux, low sugar uptake and consequent impairment in glycolysis (as we had shown earlier; **Fig. 4** of the revised manuscript) might be responsible for the reduction in *de novo* fatty acid synthesis.

As the qRT-PCR analyses were done on the RNAs isolated from the PCs and cardiac tubes to understand PC-specific lipid status, we resorted to staining the tissues with some specific dyes.

- a) Enrichment of neutral lipids in the PCs of HSD-fed flies was evident upon BODIPY and LipidTOX stainings (**Fig. 5C-E and S3A-C**).
- b) Staining the tissues with Nile blue A (staining FFA; Boumelhem et al., 2022) revealed that the PCs of HSD-fed flies are characterized by elevated levels of FFA (**Figures 5G-I**).

In the original version of the manuscript, we demonstrated, both by reporter gene expression studies and qRT-PCR analyses, that the PCs of the HSD-fed flies exhibit increased FAO, and the elevated levels of *upd3* expression in these cells depend on FAO. Therefore, we were intrigued to check whether modulating the levels of FFA generation in the PCs impacts *upd3* expression. Two sets of experiments were performed to test this proposition.

- a) In *Drosophila*, lipids are transported in hemolymph as lipoprotein particles, the most abundant being lipophorin. About 95% of all hemolymph lipids are carried by lipophorin particles, predominantly in the form of DAG. The uptake of neutral lipids from the circulating glycoprotein is mediated by the Lipophorin receptors, LpR1 and LpR2. Independently knocking down the expressions of *LpR1* and *LpR2* resulted in a detectable drop in the levels of neutral lipids in the PCs of HSD-fed flies (**Fig. 6A-E**) and a consequent reduction in FFA levels (**Fig. 6F-J**). Importantly, the increased levels of *upd3* transcripts observed in the PCs of HSD-fed flies were partially suppressed upon downregulation of either *LpR1* or *LpR 2* (**Fig. 6K**). These results demonstrate that impairing lipid uptake in the PCs of HSD-fed flies impedes the formation of lipid droplets and free fatty acids, eventually leading to partial restoration of the increased levels of *upd3* expression.
- b) In a converse set of experiments, we overexpressed *Brummer lipase (bmm)* in the PCs of HSD-fed flies to increase the levels of free fatty acids by enhanced lipolysis. Overexpression of *bmm* led to a remarkable reduction in the levels of neutral lipids in the PCs of HSD-fed flies (Figs. 6L-N) and a concomitant increase in the levels of free fatty acids (**Figs. 6P-R**). More importantly, the elevated levels of *upd3* expression associated with the PCs of HSD-fed flies got further augmented due to overexpression of *bmm* (**Fig. 6S**). Therefore, these results demonstrate that high dietary sugar-induced increased lipid uptake and its mobilization to free fatty acids in the PCs contribute to upregulating *upd3* expression.

- Express ATGL/Bmm lipase in the PC on HSD and see whether the ROS level increases? Or does increase lipolysis generate more FFA for FAO?

As suggested by the Reviewer, we expressed Bmm lipase specifically in the PCs to further enhance lipolysis. This genetic manipulation led to increased mobilization of neutral lipids (lipid droplets) (**Figs. 6L-O**) to generate more FFA (**Figs. 6P-R**) for FAO. We observed a further increase in the levels of *upd3* expression (**Fig. 6S**) in the PCs of HSD-fed flies.

We also checked whether the mobilization of accumulated lipid droplets into FFA by overexpression of *bmm* leads to any restoration of the reduced levels of ROS. We could not find any rescue in the ROS levels of the PCs of HSD-fed flies upon enhancing lipolysis (please see the figure below).

Therefore, in all likelihood, the drop in ROS is not due to an increase in TAG levels.

The results of these experiments provide a better understanding of the altered metabolic status in the PCs of HSD-fed flies and how it influences *upd3* expression in these cells. The development of insulin resistivity due to high dietary sugar impedes sugar uptake by the PCs, resulting in a consequent drop in glycolytic flux. Under this situation, the PCs demonstrate increased uptake of neutral lipids from circulating lipophorins and convert them into TAGS by lipogenesis. Since the lipophorin particles carry the lipids predominantly in the form of DAG, compared to other genes involved in lipogenesis, we see a robust increase in the expression of *Dgat2* responsible for converting DAG to TAG. BODIPY and LipidTOX stainings further endorse the presence of a large number of lipid droplets in these cells. Importantly, these PCs also demonstrate an enhanced rate of TAG mobilization into FFA. Apart from an increase in the transcript levels of the lipolytic genes (*bmm*, *Hsl*), high levels of FFA are detected in these cells. Eventually, the increased levels of FFA fuel FAO to generate energy. Most intriguingly, the elevated levels of FFA upregulate *upd3* expression in the PCs of the HSD-fed flies.

In the revised version of the manuscript, we have included these results under two separate subheadings (**Pericardial cells of HSD-fed flies exhibit altered lipid metabolism** and **Altering lipid uptake and increasing lipolysis impacts high levels of *upd3* expression in the PCs of HSD-fed flies**) in the Result section.

The inclusion of these results has enriched the quality of our manuscript. We sincerely thank the Reviewer for her/his suggestions and comments!

4. The author's claim that FAO activates HDAC-dependent transcription is not well supported by results presented in Figure 5 C-D', F-G' and the qPCR data in 5C (due to technical concerns raised in #1 and #2 points above). Further evidence is required to bolster this aspect of the study.

We have the following observations to establish the genetic connection between increased FAO and HAT-dependent transcriptional upregulation of *upd3* expression

in the PCs of the HSD-fed flies. Some of these results were presented in the original, and the others were obtained from the experiments performed during revision.

- a) Elevated levels of expression of the two HAT genes *gcn5* and *chm* under high dietary sugar conditions.
- b) Increase in the levels of H3K9ac in the PCs of HSD-fed flies.
- c) PC-specific loss of *gcn5* and *chm* results in a drop in the upregulated *upd3* expression associated with the PCs of HSD-fed flies.
- d) Downregulation of *AcCoAS* (the gene coding for the enzyme Acetyl Coenzyme A synthase that catalyzes the conversion of free acetate into acetyl coenzyme A (acetyl-CoA) and, in doing so, influences gene expression by regulating the acetylation of various proteins including histones) in the PCs of HSD-fed flies suppresses the elevated levels of *upd3* expression.
- e) The increased levels of H3K9ac in the PCs of HSD-fed flies get suppressed due to PC-specific loss of *whd*.
- f) Likewise, attenuating JNK signaling in the PCs of HSD-fed flies rescues the elevated levels of H3K9ac.
- g) Independently downregulating *whd* expression (either by KD or KO) and attenuation of JNK signaling in the PCs of HSD-fed flies suppresses the elevated levels of *upd3*.

Together, these results indicate that the upregulation in *upd3* expression observed in the PCS of HSD-fed flies is mediated by increased histone acetylation.

May 14, 2024

RE: JCB Manuscript #202306087R

Dr. Sudip Mandal
Indian Institute of Science Education and Research Mohali
Biological Sciences
IISER Mohali
Mohali
Mohali, Punjab 140306
India

Dear Dr. Mandal:

Thank you for submitting your revised manuscript entitled "High Sugar Diet-Induced Fatty Acid Oxidation Potentiates Cytokine-dependent cardiac ECM remodeling". Our apologies for the delay in providing you with this decision and thanks for your patience. As you will see, your study has been reassessed by Reviewer 1 who is supportive of publication, Reviewer 2 was not however available to re-review. We appreciate that you have put in a substantial effort in addressing the original issues and we would be happy to publish your paper in JCB pending final revisions necessary to meet our formatting guidelines (see details below).

A. MANUSCRIPT ORGANIZATION AND FORMATTING:

Full guidelines are available on our Instructions for Authors page, <https://jcb.rupress.org/submission-guidelines#revised>.
Submission of a paper that does not conform to JCB guidelines will delay the acceptance of your manuscript.

- 1) Text limits: Character count for Articles is < 40,000, not including spaces. Count includes abstract, introduction, results, discussion, and acknowledgments. Count does not include title page, figure legends, materials and methods, references, tables, or supplemental legends.
- 2) Figures limits: Articles may have up to 10 main text figures.
- 3) Figure formatting: Scale bars must be present on all microscopy images, including inset magnifications. Molecular weight or nucleic acid size markers must be included on all gel electrophoresis. In order to accommodate readers with red-green color blindness, we recommend that you avoid the use of red/green in images and graphs.
- 4) Statistical analysis: Error bars on graphic representations of numerical data must be clearly described in the figure legend. The number of independent data points (n) represented in a graph must be indicated in the legend. Statistical methods should be explained in full in the materials and methods. For figures presenting pooled data the statistical measure should be defined in the figure legends. Please also be sure to indicate the statistical tests used in each of your experiments (either in the figure legend itself or in a separate methods section) as well as the parameters of the test (for example, if you ran a t-test, please indicate if it was one- or two-sided, etc.). Also, if you used parametric tests, please indicate if the data distribution was tested for normality (and if so, how). If not, you must state something to the effect that "Data distribution was assumed to be normal but this was not formally tested."
- 5) Abstract and title: The abstract should be no longer than 160 words and should communicate the significance of the paper for a general audience. The title should be less than 100 characters including spaces. Make the title concise but accessible to a general readership.
- 6) Materials and methods: * Should be comprehensive and not simply reference a previous publication for details on how an experiment was performed. Please provide full descriptions in the text for readers who may not have access to referenced manuscripts. *
- 7) All antibodies, cell lines, animals, and tools used in the manuscript should be described in full, including accession numbers for materials available in a public repository such as the Resource Identification Portal. Please be sure to provide the sequences for all of your primers/oligos and RNAi constructs in the materials and methods. You must also indicate in the methods the source, species, and catalog numbers (where appropriate) for all of your antibodies. Please also indicate the acquisition and quantification methods for immunoblotting/western blots.

8) Microscope image acquisition: The following information must be provided about the acquisition and processing of images:

- Make and model of microscope
- Type, magnification, and numerical aperture of the objective lenses
- Temperature
- Imaging medium
- Fluorochromes
- Camera make and model
- Acquisition software
- Any software used for image processing subsequent to data acquisition. Please include details and types of operations involved (e.g., type of deconvolution, 3D reconstitutions, surface or volume rendering, gamma adjustments, etc.).

10) Supplemental materials: There are strict limits on the allowable amount of supplemental data. Articles may have up to 5 supplemental figures. Please also note that tables, like figures, should be provided as individual, editable files. A summary of all supplemental material should appear at the end of the Materials and methods section.

13) ORCID IDs: ORCID IDs are unique identifiers allowing researchers to create a record of their various scholarly contributions in a single place. Please note that ORCID IDs are now *required* for all authors. At resubmission of your final files, please be sure to provide your ORCID ID and those of all co-authors.

Please note that JCB now requires authors to submit Source Data used to generate figures containing gels and Western blots with all revised manuscripts. This Source Data consists of fully uncropped and unprocessed images for each gel/blot displayed in the main and supplemental figures. Since your paper includes cropped gel and/or blot images, please be sure to provide one Source Data file for each figure that contains gels and/or blots along with your revised manuscript files. File names for Source Data figures should be alphanumeric without any spaces or special characters (i.e., SourceDataF#, where F# refers to the associated main figure number or SourceDataFS# for those associated with Supplementary figures). The lanes of the gels/blots should be labeled as they are in the associated figure, the place where cropping was applied should be marked (with a box), and molecular weight/size standards should be labeled wherever possible.

Journal of Cell Biology now requires a data availability statement for all research article submissions. These statements will be published in the article directly above the Acknowledgments. The statement should address all data underlying the research presented in the manuscript. Please visit the JCB instructions for authors for guidelines and examples of statements at (<https://rupress.org/jcb/pages/editorial-policies#data-availability-statement>).

B. FINAL FILES:

Thank you for your attention to these final processing requirements. Please revise and format the manuscript and upload materials within 7 days. If you need an extension for whatever reason, please let us know and we can work with you to determine a suitable revision period.

Thank you for this interesting contribution, we look forward to publishing your paper in Journal of Cell Biology.

Sincerely,

Mark Peifer
Monitoring Editor

Andrea L. Marat
Senior Scientific Editor

Journal of Cell Biology

Reviewer #1 (Comments to the Authors (Required)):

The authors addressed all of my previous concerns and have included a significant amount of new data that help clarify several key aspects of their study.